# Connexin26 mediates $CO_2$-dependent regulation of breathing via glial cells of the medulla oblongata

Joseph van de Wiel [1], Louise Meigh[1], Amol Bhandare [1], Jonathan Cook [1], Sarbjit Nijjar[1], Robert Huckstepp [1] & Nicholas Dale [1✉]

Breathing is highly sensitive to the $PCO_2$ of arterial blood. Although $CO_2$ is detected via the proxy of pH, $CO_2$ acting directly via Cx26 may also contribute to the regulation of breathing. Here we exploit our knowledge of the structural motif of $CO_2$-binding to Cx26 to devise a dominant negative subunit (Cx26[DN]) that removes the $CO_2$-sensitivity from endogenously expressed wild type Cx26. Expression of Cx26[DN] in glial cells of a circumscribed region of the mouse medulla - the caudal parapyramidal area – reduced the adaptive change in tidal volume and minute ventilation by approximately 30% at 6% inspired $CO_2$. As central chemosensors mediate about 70% of the total response to hypercapnia, $CO_2$-sensing via Cx26 in the caudal parapyramidal area contributed about 45% of the centrally-mediated ventilatory response to $CO_2$. Our data unequivocally link the direct sensing of $CO_2$ to the chemosensory control of breathing and demonstrates that $CO_2$-binding to Cx26 is a key transduction step in this fundamental process.

[1] School of Life Sciences, University of Warwick, Coventry CV4 7AL, UK. ✉email: N.E.Dale@warwick.ac.uk

Breathing is a vital function that maintains the partial pressures of $O_2$ and $CO_2$ in arterial blood within the physiological limits. Chemosensory reflexes regulate the frequency and depth of breathing to ensure homoeostatic control of blood gases[1]. Historically, the ventral surface of the medulla oblongata has been recognised as an important location of central respiratory chemosensors[2–6]. Recent work has focussed on two populations of neurons thought to contribute to the chemosensory control of breathing: those of the retrotrapezoid nucleus (RTN)[7–11] and the medullary raphé[12–16].

According to traditional consensus, $CO_2$ is detected via the consequent change in pH, and pH is a sufficient stimulus for all adaptive changes in breathing in response to hypercapnia[6]. pH-sensitive $K^+$ channels (TASKs and KIRs) are potential transducers. Although TASK-1 in the peripheral chemosensors of the carotid body (CB) contributes to overall pH/$CO_2$ chemosensitivity[17], TASK-1 does not appear to play a role in central pH/$CO_2$ chemosensing[18]. By contrast, TASK-2 may act as a central sensor of pH and contribute to adaptive changes in breathing[11,19]. Recently, a pH sensitive receptor, GPR4, has been linked to central chemosensitivity in the RTN. Complete deletion of this gene (from all tissues) greatly reduced the $CO_2$ chemosensitivity in mice[11]. However, GPR4 is widely expressed in neurons including those of the medullary raphé and peripheral chemosensors, as well as the endothelium[20]. Moreover, systemic injection of a selective GPR4 antagonist modestly reduced the ventilatory response to $CO_2$, but this same antagonist when administered centrally had no effect on the $CO_2$ sensitivity of breathing[20]. A mechanism of pH-dependent release of ATP from ventral medullary glial cells may also contribute to the $CO_2$-dependent regulation of breathing[21].

There is considerable evidence that $CO_2$ can have additional independent effects from pH on central respiratory chemosensors[22–24]. Connexin26 (Cx26) is known to be present at the ventral medullary surface in the caudal and rostral chemosensory areas[25,26]. We have shown that $CO_2$ directly binds to Cx26 hemichannels and causes them to open[27,28]. We have identified the critical amino acid residues that are necessary and sufficient for this process[28,29]. Cx26 hemichannels can be gated by a number of stimuli. They are opened by voltage at potentials > $-20$ mV[30] and closed by acidification[31]. Like all connexins, Cx26 hemichannels can be opened by removal of extracellular $Ca^{2+}$[32]. However, the $CO_2$-dependent opening of Cx26 hemichannels can occur in the absence of membrane depolarisation and at physiological levels of extracellular $Ca^{2+}$[27–29,33].

This direct gating of Cx26 is an important mechanism that underlies $CO_2$-dependent ATP release[26,27,34,35] and provides a potential mechanism for the direct action of $CO_2$ on breathing. Nevertheless, the role of direct sensing of $CO_2$ in the regulation of breathing remains uncertain because genetic evidence linking Cx26 to the control of breathing has been lacking, and the cells that could mediate direct $CO_2$ sensing via Cx26 have not been identified.

In this study, we have addressed both of these issues by exploiting our knowledge of the binding of $CO_2$ to Cx26 to devise a dominant-negative subunit (Cx26$^{DN}$) that removes $CO_2$-sensitivity from endogenous wild-type (WT) Cx26 hemichannels. By using a lentiviral construct to drive the expression of Cx26$^{DN}$ in glial cells of the ventral medulla, we have obtained evidence that links $CO_2$-dependent modulation of Cx26 in glial cells present in a small circumscribed area of the ventral medulla to the adaptive control of breathing.

## Results

### Rationale for the design of Cx26$^{DN}$. $CO_2$ causes Cx26 hemichannels to open via carbamylation of K125 and subsequent

formation of a salt bridge between the carbamylated lysine side chain and R104 of the neighbouring subunit (Fig. 1a)[28]. As Cx26 hemichannels are hexameric, there are potentially six binding sites for $CO_2$ suggesting the potential for highly cooperative binding of $CO_2$. Indeed, Cx26 is steeply sensitive to changes in $PCO_2$ around its physiological level of about 40 mmHg (Fig. 1b)[27]. We reasoned that introducing two mutations: K125R to prevent $CO_2$-dependent carbamylation and R104A to prevent salt bridge formation ("carbamate bridges") to a neighbouring carbamylated subunit, should produce a dominant-negative subunit (Fig. 1c). If such a subunit coassembled into the Cx26 hexamer, it would likely have a dominant-negative action as it would remove the capacity to form at least two out of the six possible carbamate bridges (Fig. 1c).

**Cx26$^{DN}$ coassembles with Cx26$^{WT}$.** To determine whether the dominant-negative subunit will effectively coassemble into a hexamer with the WT subunit, we used acceptor depletion Förster resonance energy transfer (FRET; Fig. 2). We exploited the Clover-mRuby2 FRET pair[36] by tagging Cx26$^{WT}$ and Cx26$^{DN}$ with Clover (donor) and mRuby2 (acceptor). When mRuby2 was bleached in HeLa cells coexpressing either Cx26$^{WT}$-Clover and Cx26$^{WT}$-mRuby2 or Cx26$^{WT}$-Clover and Cx26$^{DN}$-mRuby2, the fluorescence of the Clover was enhanced where these two fluorophores were colocalised (Fig. 2). The FRET efficiency for Cx26$^{WT}$-Clover and Cx26$^{WT}$-mRuby2 or Cx26$^{WT}$-Clover and Cx26$^{DN}$-mRuby2 was very similar (Fig. 3). This suggests that the Cx26$^{DN}$ subunit interacts closely with Cx26$^{WT}$. In principle, this could be because a homomeric hexamer of Cx26$^{WT}$ was sufficiently close to a homomeric hexamer of Cx26$^{DN}$ by random association in the membrane rather than assembly of a heteromeric hexamers comprised of both types of subunits. We

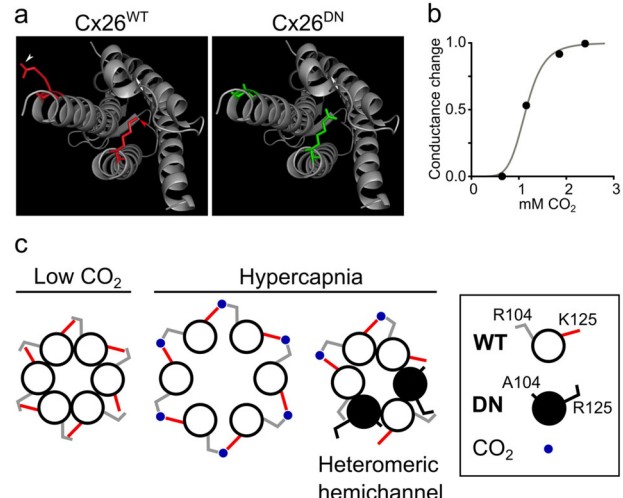

**Fig. 1 Rationale for the design of a dominant-negative Cx26 subunit (Cx26$^{DN}$) to inhibit $CO_2$-mediated hemichannel opening. a** Ribbon diagram, showing a single connexin26 subunit, highlighting the two amino acid residues vital for $CO_2$ sensing—R104 (white arrowhead) and K125 (red arrowhead, Cx26$^{WT}$, left). In the Cx26$^{DN}$ mutant, these two residues are mutated—R104A and K125R (Cx26$^{DN}$, right). This will prevent carbamylation of the subunit and formation of a salt "carbamate" bridge with adjacent connexin subunit in the hexamer. **b** $CO_2$-dependent whole-cell clamp conductance changes from HeLa cells stably expressing Cx26$^{WT}$ (data from ref. [27]). The grey line is drawn with a Hill coefficient of 6, suggesting that hemichannel opening to $CO_2$ is highly cooperative. **c** Hypothesised coassembly of Cx26$^{WT}$ and Cx26$^{DN}$ into heteromeric hemichannels that will be insensitive to $CO_2$ because insufficient carbamate bridge formation will occur to induced channel opening.

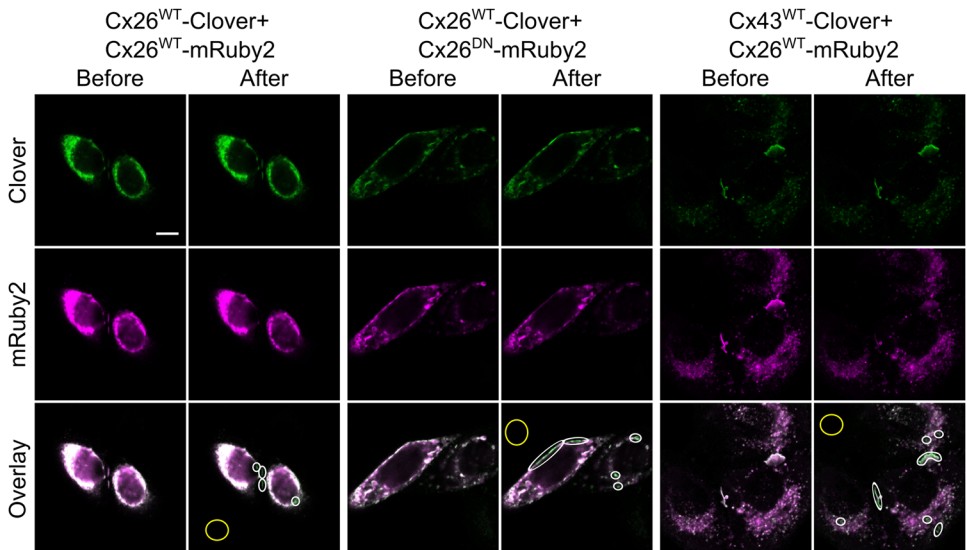

**Fig. 2 FRET signal between connexin variants.** Example images of acceptor depletion Förster resonance energy transfer (ad-FRET) experiments. HeLa cells were co-transfected with equal amounts of DNA transcripts (to express connexin–fluorophore constructs) and PFA fixed after 48 h. Two channels were recorded: 495–545 nm (Clover, green) and 650–700 nm (mRuby2, magenta), and images were acquired sequentially with 458- and 561-nm argon lasers, respectively. Photobleaching was performed using the 561-nm laser for 80 frames at 100% power, targeting ROIs. Three combinations of connexin–Clover (donor, green) and connexin–mRuby2 (acceptor, red) are shown before and after bleaching. Colocalisation is shown in white in the overlay images, along with bleached areas (white ovals) and background references (yellow ovals). Following acceptor bleaching, ROIs in Cx26$^{WT}$-Clover + Cx26$^{WT}$-mRuby2 and Cx26$^{WT}$-Clover + Cx26$^{DN}$-mRuby2 samples show enhanced fluorescence intensity of donor (Clover, green) and reduced fluorescence intensity of acceptor (mRuby2, magenta). ROIs in Cx43$^{WT}$-Clover + Cx26$^{WT}$-mRuby2 samples show almost no change in fluorescence intensity of donor (Clover, green) and reduced fluorescence intensity of acceptor (mRuby2, magenta). Colocalisation measurements decrease mainly due to photobleaching of mRuby2 and subsequent elimination of its fluorescence. All co-transfection combinations showed statistically significant colocalisation at the highest resolution for the images. Colocalisation threshold calculations were carried out in ImageJ using the Costes method; 100 iterations, omitting zero–zero pixels in threshold calculation. Statistical significance was calculated for entire images and individual ROIs, with no difference between either calculation ($p$ value = 1 for all images, with a significance threshold of $p > 0.95$). Scale bar, 10 μm.

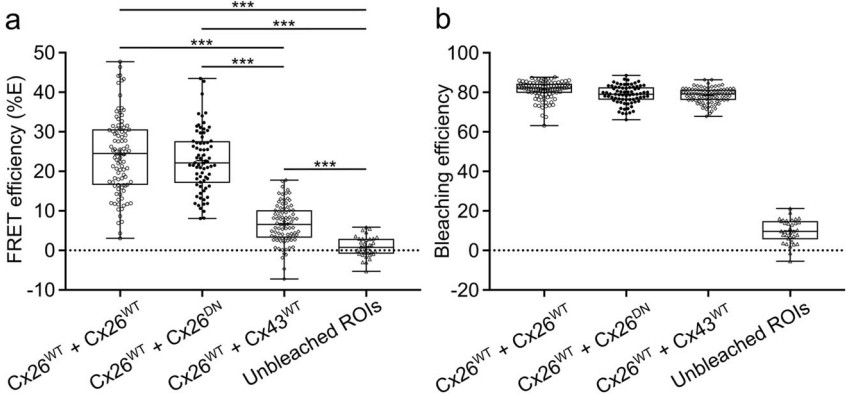

**Fig. 3 FRET efficiency of coexpressed connexin variants. a** Box and whisker plots showing the difference in FRET efficiency (%$E$) across different connexin co-expression connexin samples. FRET efficiency was calculated from background-adjusted ROIs as: %$E = 100 × (\text{clover}_{post} − \text{clover}_{pre})/\text{clover}_{post}$. One-way ANOVA was carried out in SPSS. Post hoc testing revealed all individual comparisons to be significant at ***$p < 0.001$, with the exception of the comparison between Cx26$^{WT}$ + Cx26$^{WT}$ and Cx26$^{WT}$ + Cx26$^{DN}$ data sets, which were not significantly different. Each dot represents a different ROI. **b** Box and whisker plot showing mRuby2 bleaching efficiency during the acceptor depletion step. While a small amount of bleaching occurred in untargeted regions (presumably due to light scattering and/or reflection), targeted ROIs received drastically greater bleaching. Importantly, all targeted regions showed highly similar bleaching efficiencies. Bleaching efficiency was calculated from background-adjusted ROIs as: Bleaching = $(1 − (\text{mRuby2}_{post}/\text{mRuby2}_{pre}))$. Boxes show the interquartile range, the median is indicated by the horizontal line within the box, and the mean is indicated by the cross within the box. Range bars show minimum and maximum values.

therefore also looked at FRET interactions between Cx43$^{WT}$ and Cx26$^{WT}$. These two connexin subunits do not form heteromeric hexamers, but homomeric hexamers of the two types could come close to each other in the plasma membrane by random association. We found that the FRET efficiency of this interaction was much lower than that of Cx26$^{WT}$-Cx26$^{WT}$ or Cx26$^{WT}$-Cx26$^{DN}$

(Figs. 2 and 3). This suggests that Cx26$^{DN}$ does indeed coassemble into a hexamer with Cx26$^{WT}$. This hypothesis gains further support from our analysis which shows that the FRET efficiency for Cx26$^{WT}$-Cx26$^{DN}$ is negatively correlated with the donor-to-acceptor (D-A) ratio (Supplementary Fig. 1), which is indicative of assembly into heteromeric hemichannels[37]. By

contrast, the FRET efficiency for $Cx43^{WT}$-$Cx26^{WT}$ shows no such correlation with the D-A ratio (Supplementary Fig. 1). There is a positive correlation of FRET efficiency with acceptor level, which may indicate some additional random association of homomeric connexons comprised of $Cx26^{WT}$ and $Cx26^{DN}$[37] (Supplementary Fig. 1). Nevertheless, our data strongly indicate that the $Cx26^{DN}$ subunit effectively coassembles with $Cx26^{WT}$ subunits to form heteromeric hemichannels.

**$Cx26^{DN}$ has a dominant-negative action on the $CO_2$ sensitivity of $Cx26^{WT}$.** We have previously shown that hemichannels comprised of $Cx26^{K125R}$ and $Cx26^{R104A}$ are insensitive to $CO_2$[28]. We therefore confirmed that $Cx26^{DN}$ hemichannels, which carry both of these mutations, are also insensitive to $CO_2$ (Supplementary Fig. 2). This important observation shows that if $Cx26^{DN}$ were to form homomeric hemichannels in vivo they would have no impact on the $CO_2$ sensitivity of cells expressing them.

To assess the capacity of $Cx26^{DN}$ to act as a dominant-negative construct with respect to $CO_2$ sensitivity, we transfected this subunit into HeLa cells that stably expressed $Cx26^{WT}$. We then used a dye loading assay to assess how the sensitivity of these HeLa cells to $CO_2$ changed with time. Four days after transfection, the HeLa cells that coexpressed $Cx26^{DN}$ with $Cx26^{WT}$ were as sensitive to $CO_2$ as those HeLa cells that only expressed $Cx26^{WT}$ (Fig. 4). However, 6 days after transfection, the HeLa cells that coexpressed $Cx26^{DN}$ were insensitive to $CO_2$ (Fig. 4). We conclude that $Cx26^{DN}$ has a dominant-negative effect on the $CO_2$ sensitivity of $Cx26^{WT}$ and exerts this effect by coassembling into heteromeric hexamers with $Cx26^{WT}$ in the manner hypothesised in Fig. 1.

**In vivo expression of $Cx26^{DN}$ blunts the $CO_2$ sensitivity of breathing.** The $Cx26^{DN}$ subunit has the potential to be a genetic tool that could remove $CO_2$ sensitivity from Cx26 and thus probe this aspect of Cx26 function without deleting the Cx26 gene. This has the advantage of leaving other signalling roles of Cx26 intact and also of linking any $CO_2$-dependent physiological functions to the $CO_2$-binding site of Cx26 itself. We therefore created a lentiviral construct that contained either $Cx26^{DN}$ or $Cx26^{WT}$ (a control for the $Cx26^{DN}$ construct) under the control of a bidirectional cell-specific glial fibrillary acidic protein (GFAP) promoter[38] (Supplementary Fig. 3a). Rather than tagging a fluorescent protein to the C-terminus of the Cx26 variants, we expressed Clover behind an internal ribosome entry site (IRES) so that the cells that expressed the Cx26 variants could be identified and quantified (Supplementary Fig. 3a). Stereotaxic injection of the lentiviral construct into the ventral medulla oblongata showed that lentiviral construct drove expression of the Cx26 variants selectively in GFAP+ cells (Supplementary Fig. 3b).

We used bilateral stereotaxic injections of the lentiviral constructs into the medulla oblongata to assess the effect of $Cx26^{DN}$ on the $CO_2$ sensitivity of breathing by means of whole-body plethysmography (Supplementary Fig. 4). Pilot experiments suggested that transduction of glial cells at the ventral medullary surface in a region ventral and medial to the lateral reticular nucleus reduced the adaptive changes in breathing to hypercapnia 3 weeks after injection of the virus (Supplementary Fig. 5). From this pilot work, we designed a study to achieve a statistical power of 0.8 at a significance level of 0.05 in which we injected 12 mice with $Cx26^{WT}$ and 12 mice with $Cx26^{DN}$ and followed how the $CO_2$ sensitivity of breathing changed with time following the lentiviral injection (Fig. 5a). Two weeks after viral transduction, the change in tidal volume to 6% $CO_2$ in the $Cx26^{DN}$-injected mice was less than that of the $Cx26^{WT}$-injected mice (two-way mixed-effects analysis of variance (ANOVA), $p = 0.008$; post hoc

one-tailed $t$ test, $p = 0.002$). The change in minute ventilation was also less in the $Cx26^{DN}$-injected mice compared to the $Cx26^{WT}$-injected mice (two-way mixed-effects ANOVA, $p = 0.015$; post hoc one-tailed $t$ test, $p = 0.007$). There was no readily discernible difference in the changes in respiratory frequency to $CO_2$ between the $Cx26^{WT}$- and $Cx26^{DN}$-injected mice. Three weeks following the viral injection, these differences had disappeared, presumably due to some compensatory mechanism within the respiratory networks[39,40]. Following the pilot study, we were able to place our injection more medially into the caudal parapyramidal area. More accurately targeting the injection of virus particles to the correct area likely led to the quicker onset of the phenotype, as transfection of the relevant cells would thus have been more efficient and rapid.

Post hoc tissue staining confirmed the location of the transduced glial cells and highlighted cells at the very surface of the ventral medulla that had long processes projecting rostrally and dorsally into the parenchyma of the medulla (Fig. 5b). The difference between $Cx26^{WT}$- and $Cx26^{DN}$-transduced mice could not be attributed to the increased expression of $Cx26^{WT}$ and hence $CO_2$ sensitivity, as sham operated mice (performed as part of the same study) showed no difference in $CO_2$ sensitivity from the $Cx26^{WT}$ mice (Supplementary Fig. 7). In both the pilot and main experiments, the effect of $Cx26^{DN}$ was to alter the relationship of tidal volume and minute ventilation vs inspired $CO_2$: specifically, $Cx26^{DN}$ reduced the increase in both of these parameters that occurs at 6% inspired $CO_2$ by ~30% compared to the control ($Cx26^{WT}$, see Supplementary Figs. 5 and 6 and Supplementary Tables 1 and 2).

Expression of $Cx26^{DN}$ in glial cells in this location ventral and medial to the lateral reticular nucleus appeared to be uniquely able to alter the $CO_2$ sensitivity of breathing. Transduction of glial cells in the RTN (Supplementary Fig. 8) or even more caudally in the medulla (Supplementary Fig. 9) had no effect on the $CO_2$ sensitivity of breathing. We therefore conclude that Cx26 in a circumscribed population of GFAP+ cells ventral and medial to the lateral reticular nucleus contributes to the $CO_2$-dependent regulation of breathing. To our knowledge, this is the first mechanistic evidence that shows a direct effect of $CO_2$ on breathing and links the structural biology of $CO_2$ binding to Cx26 to the regulation of breathing. While the lentiviral construct transduced typical astrocytes (e.g., Fig. 5b), it consistently transduced glial cells that had a soma at the very ventral edge of the medulla. These glial cells were unusual in that they had very long processes, which extended rostrally (Figs. 5b and 6) and also medially (Fig. 6).

## Discussion

By devising a dominant-negative subunit $Cx26^{DN}$, which coassembles with endogenously expressed $Cx26^{WT}$ to remove $CO_2$ sensitivity from the resulting heteromeric hemichannels, we have demonstrated a clear link between Cx26-mediated $CO_2$ sensing and the regulation of breathing. Our study also links the structural motif of $CO_2$ binding in Cx26—the carbamylation motif—to the $CO_2$-dependent regulation of breathing. $CO_2$ carbamylation happens spontaneously and was originally described as the basis of the $CO_2$ Bohr effect[41]. Carbamylation of lysine residues has also been established in RuBisco[42], a key enzyme for photosynthetic carbon fixation, and in microbial beta-lactamases[43,44]. $CO_2$-dependent carbamylation as a general and important post-translational protein modification involved in physiological regulation was proposed by George Lorimer[45]. It is clear from the known examples, and systematic application of mass spectrometric tools[46], that only specific lysine residues in some proteins are able to be carbamylated. Our data now suggest that $CO_2$

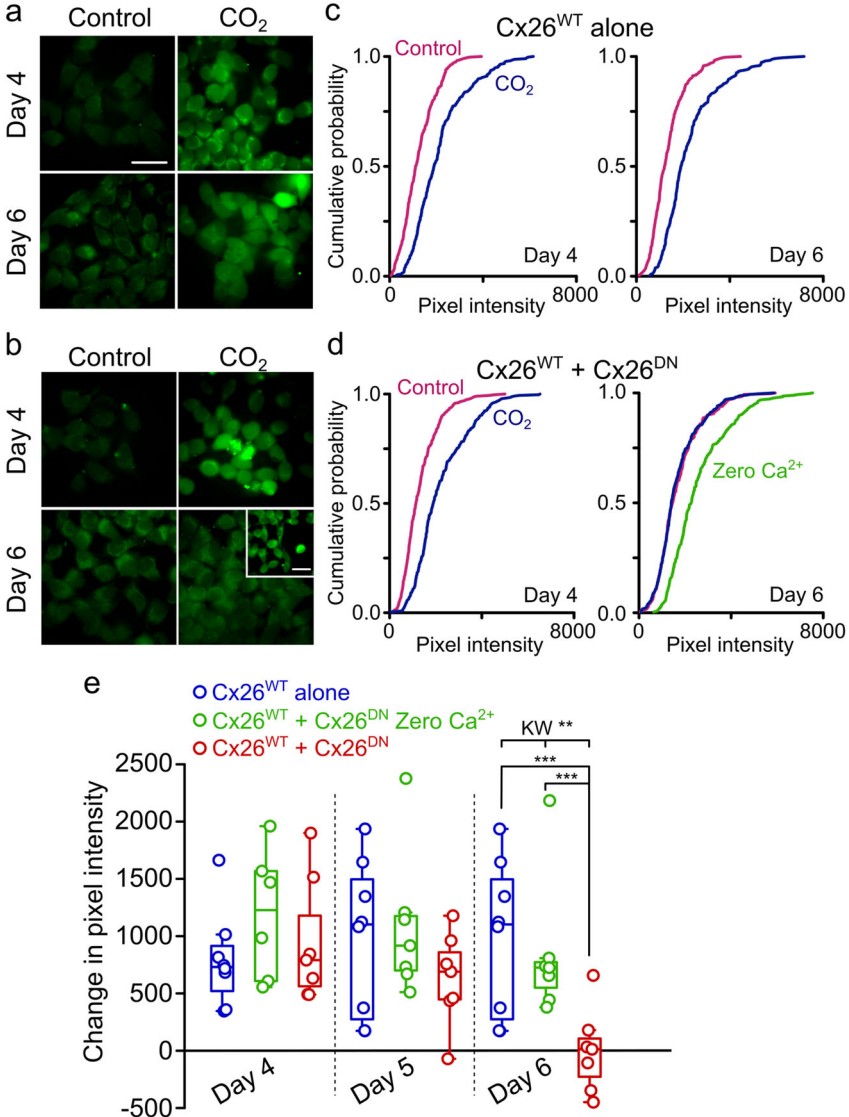

**Fig. 4 Cx26$^{DN}$ removes CO$_2$ sensitivity in HeLa cells stably expressing Cx26$^{WT}$.** Dye loading under 35 mmHg PCO$_2$ (control) or 55 mmHg PCO$_2$ (hypercapnic) conditions revealed how the CO$_2$ sensitivity of HeLa cells stably expressing Cx26 (Cx26-HeLa cells) changes over time after transfection with Cx26$^{DN}$. **a, b** Representative dye-loading images of Cx26-HeLa cells at day 4 and day 6 days after cells were either untreated **a** or transfected with Cx26$^{DN}$ **b**. In **b**, the inset represents a Zero Ca$^{2+}$ control to demonstrate the presence of functional hemichannels even when the HeLa cells showed no CO$_2$-dependent dye loading. **c, d** Cumulative probability distributions comparing mean pixel intensity for each condition at day 4 and day 6 (untreated Cx26-HeLa cells, **c**; Cx26-HeLa cells transfected with Cx26$^{DN}$, **d**; $n > 40$ cells per treatment repeat, with at least 5 independent repeats for each treatment). The cumulative distributions show every data point (cell fluorescence intensity measurement). **e** Median change in pixel intensity caused by 55 mmHg PCO$_2$ and Zero Ca$^{2+}$ from baseline (35 mmHg) over days 4, 5, and 6 post-transfection. At day 6, median pixel intensities (from 7 independent repeats) were compared using the Kruskal–Wallis ANOVA ($\chi^2 = 9.85$, df $= 2$, $p = 0.007$**) and post hoc with Mann–Whitney $U$ test (Cx26$^{WT}$ vs Cx26$^{WT}$ +Cx26$^{DN}$, $W = 51$, $p = 0.003$***; Cx26$^{WT}$+Cx26$^{DN}$ Zero Ca$^{2+}$ vs Cx26$^{WT}$+Cx26$^{DN}$, $W = 46$, $p = 0.002$***). Each circle represents one independent replication (independent transfections and cell cultures). Boxes show the interquartile range, the median is indicated by the horizontal line within the box, and the whisker is 1.5 times the interquartile range. Scale bars, 40 µm.

carbamylation plays an important physiological role in the control of breathing.

The direct and independent role of CO$_2$ sensing in central chemoreception was first suggested by Shams[23]. The data in this paper provide unequivocal evidence to demonstrate a molecular mechanism and role for direct CO$_2$ sensing in the control of breathing. This molecular mechanism and pathway functions independently from any secondary changes in pH that could result from the altered balance between CO$_2$ and HCO$_3^-$ during hypercapnia. Our data further suggest that the role of Cx26 and direct CO$_2$ sensing is restricted to an area of the ventral medullary surface in the caudal brain stem—ventral and medial to the lateral

reticular nucleus. This area corresponds to the classically described caudal chemosensing area[3,4,47] and in particular a subregion that has been more recently studied and termed the "caudal parapyramidal" area. The caudal parapyramidal area contains serotonergic neurons that are highly pH sensitive[48,49]. Interestingly, our lentiviral vector consistently transduced glial cells at the very ventral surface of the medulla in this region (Fig. 7). Our previous work has shown, with the aid of a knock-in reporter, that Cx26 is expressed in GFAP+ cells with cell bodies at the very surface of the parenchyma (Fig. 10 of ref. [26]). Our current work is consistent with this finding and shows that these superficial GFAP+ cells have a cell body with a flattened edge that forms

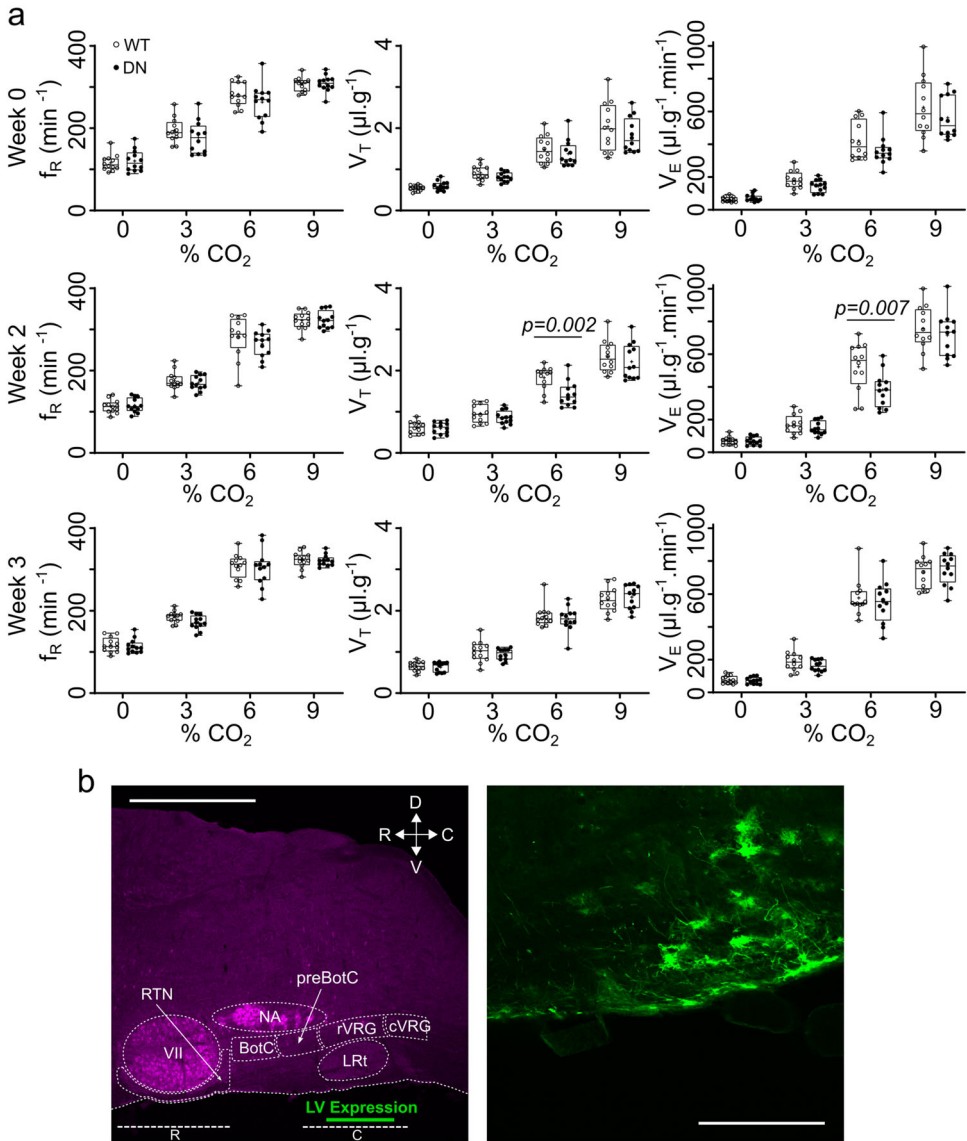

**Fig. 5 Connexin26-mediated hypercapnic breathing response in conscious mice. a** Mice aged 11–14 weeks were bilaterally injected with lentivirus (LV) at the ventral medullary surface (VMS) to introduce either the $Cx26^{DN}$ or $Cx26^{WT}$ gene under the control of a GFAP promoter into genomic DNA. Whole-body plethysmographic measurements of frequency ($f_R$), tidal volume ($V_T$), and minute ventilation ($V_E$) were recorded for each mouse at 0, 3, 6, and 9% $CO_2$, before (week 0) and after (weeks 2 and 3) LV transduction. Two weeks after transduction, there was a difference (two-way mixed-effects ANOVA followed by post hoc $t$ test: $V_T$ $F = 4.245$, df $= 3$, $p = 0.008$; $V_E$ $F = 3.738$, df $= 3$, $p = 0.015$; $p$ values on figure given for post hoc comparisons) in adaptive changes in tidal volume to 6% $CO_2$ when comparing mice expressing $Cx26^{WT}$ (empty circles, $n = 12$ mice) and mice expressing $Cx26^{DN}$ (black circles, $n = 12$ mice). The median is indicated as a horizontal line within the box, and the mean is represented by a cross within the box. Range bars show minimum and maximum values. **b** Location of GFAP:Cx26 LV construct expression (green) in the sagittal plane—scale bars, 1 mm (left), 200 μm (right). R rostral chemosensitive site, C caudal chemosensitive site, NA nucleus ambiguous, VII facial nucleus, preBot preBötzinger complex, Bot Bötzinger complex, rVRG rostral ventral respiratory group, RTN retrotrapezoid nucleus, cVRG caudal ventral respiratory group, LRt lateral reticular nucleus, D dorsal, V ventral, R rostral, C caudal.

part of the very surface of the marginal glial layer and long dorsally directed processes that extended both rostrally and medially (Figs. 5b and 6). The cell body is ideally placed to detect changes in $PCO_2$ in the cerebrospinal fluid (CSF), but their processes projecting both rostrally and medially could contact neurons of the preBötzinger complex to directly alter inspiratory activity or neurons of the raphé obscurus or pallidus, which detect changes in pH and are part of the chemosensory network[50–52].

Our findings are robust and reproducible. We discovered the importance of Cx26 in the caudal parapyramidal area in a pilot experiment (Supplementary Fig. 5) and used these data to design a properly powered experiment to test the importance of Cx26 in

this area. For practical reasons, the groups of mice for the experiment reported in Fig. 5 were divided into subcohorts of six per treatment group. In each of these subcohorts, the effect of $Cx26^{DN}$ replicated that seen in the pilot experiment. We also observed the effect of $Cx26^{DN}$ at 1.5, 2, and 2.5 weeks; however, by 3 weeks post-transduction the effect of $Cx26^{DN}$ on breathing had disappeared. This is presumably due to compensation within the respiratory network, which is known to be highly plastic[39]. For example, the effects of CB denervation are largely reversed over the course of 2 weeks[53]; mice engineered to express a mutant Phox2b gene selectively in neurons of the RTN have no $CO_2$ chemosensitivity for the first 9 days postnatally but recover about

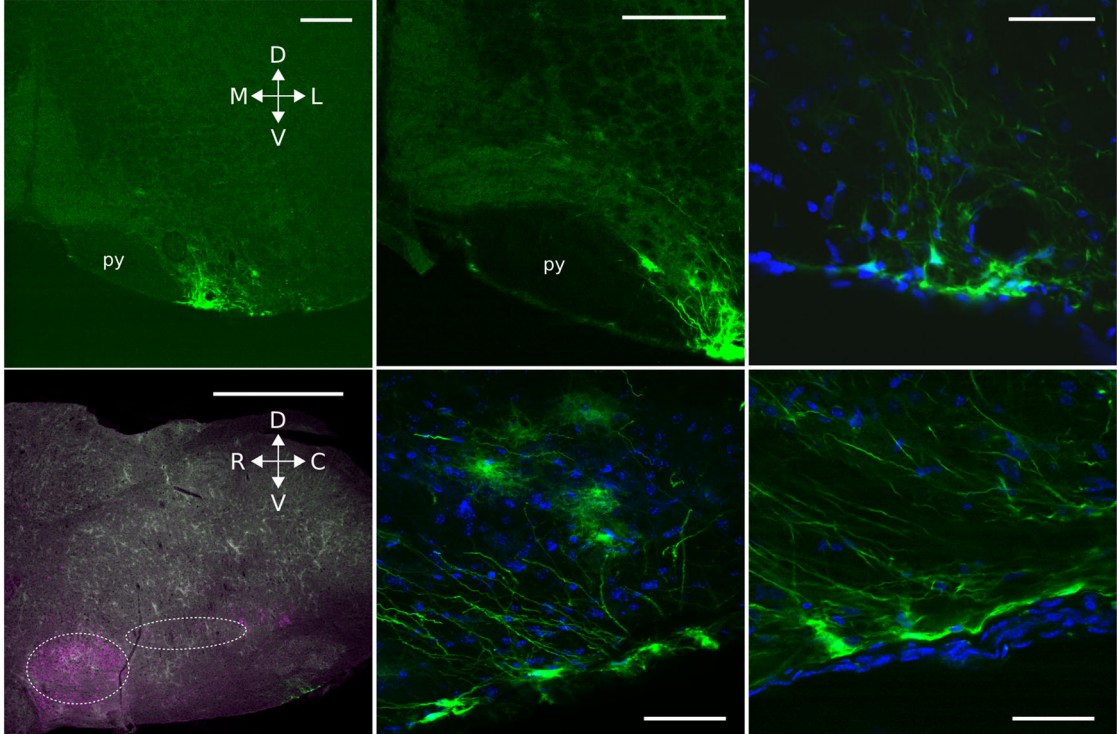

**Fig. 6 The putative chemosensory cells extend processes dorsal, medial, and rostral.** Top three images: Coronal sections. Bottom three images: parasagittal sections showing the location of the transduced cells (left) and examples of the cells at higher magnification (middle and right). Cells expressing GFAP:Cx26:Clover (green) at the ventral medullary surface have a morphology unlike that of astrocytes, with a cell body at the very margin of the ventral surface and long processes that extend deep into the brain in the direction of respiratory nuclei. The ventrolateral respiratory column lies caudal to the VII nucleus and ventral to the nucleus ambiguous (dashed ovals). Choline acetyltransferase staining (magenta). py pyramids. Scale bars: bottom-left, 1 mm; top-left and top-middle, 200 µm; top-right and bottom-middle and bottom-right, 50 µm. M medial, L lateral, D dorsal, V ventral, R rostral, C caudal.

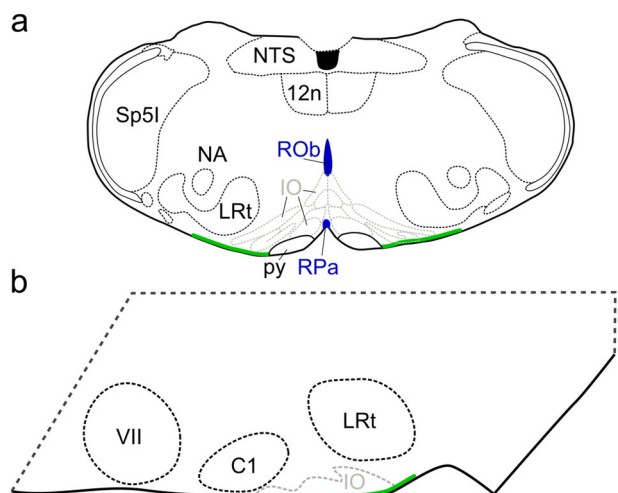

**Fig. 7 Schematic of the location of glial cells which, when transduced with Cx26$^{DN}$ at the medullary surface, reduced the chemosensitivity of breathing.** Location of cells is shown in the coronal (**a**) and parasagittal (**b**) planes. The cells are in an area ventral to the lateral reticular nucleus (LRt) and inferior olive (IO). The area reaches laterally to the same parasagittal plane as the nucleus ambiguus (NA) extends medially to the pyramids (py). NTS nucleus tractus solitarius, 12n hypoglossal nucleus, ROb raphé obscurus, RPa raphé pallidus, SP5I spinal trigeminal nucleus interpolar part, VII facial nucleus, C1 lateral paragigantocellular nucleus C1.

40% of their $CO_2$ chemosensitivity by 4 months[54]. Interestingly, the recovered chemosensitivity in these mice was mediated almost exclusively via an adaptive increase in tidal volume, rather than frequency[54]—it is therefore tempting to speculate that this could have arisen via the progressive postnatal expression of Cx26 in glial cells of the caudal parapyramidal area. There are several ways that the respiratory and chemosensory networks could compensate for the effects of Cx26$^{DN}$, the most obvious being via strengthening of other chemosensory inputs such as those from the raphé, RTN, or CBs. However, a more subtle compensatory mechanism might be through upregulation of ATP receptors on the neurons downstream from the chemosensory glia in the caudal parapyramidal area. It is unlikely that we transduced every chemosensitive glial cell in this area, hence an upregulation of ATP receptors could maximise the effects of the non-transduced glial cells that remain chemosensitive.

Overexpression of Cx26$^{WT}$ by itself did not enhance the chemosensitivity of breathing. Our construct drives the expression in GFAP+ cells, which could either be part of the chemosensory network or outside of it. For those glial cells within the chemosensory network, it may be that the endogenous levels of expression are sufficient and additional expression of Cx26 can give no further gain of chemosensitivity—a "ceiling" effect. Cx26$^{WT}$ expression in chemosensory cells outside of the network may be ineffective in enhancing chemosensitivity because these cells do not project to (and release ATP in) the correct locations to alter breathing.

Cx26$^{DN}$ and Cx26$^{WT}$ colocalise to gap junction plaques and the FRET analysis shows that they coassemble into heteromeric

gap junctions (Fig. 2). By itself, $Cx26^{DN}$ forms functional gap junctions, which allow dye transfer (Supplementary Fig. 10). $Cx26^{WT}$ gap junctions can be closed by $CO_2$ acting via the same carbamylation motif that opens the hemichannel[55] and $Cx26^{DN}$ subunits form gap junctions that are insensitive to $CO_2$ (Supplementary Fig. 10). We therefore expect that $Cx26^{DN}$ would have a similar dominant-negative action on the $CO_2$ sensitivity of endogenous Cx26 gap junctions and cannot exclude that $Cx26^{DN}$-mediated loss of $CO_2$-dependent gap junction closure could contribute to the observed results. However, there is abundant evidence for the presence of Cx26 hemichannels[26], and it is mechanistically simpler to understand how the $CO_2$-gated release of ATP via Cx26 hemichannels could result in enhanced neuronal excitation and hence the adaptive changes in ventilation, compared to $CO_2$-gated loss of gap junction communication between glial cells.

The RTN has received extensive attention as a chemosensory area[8,11,56–58]. It is interesting that $Cx26^{DN}$ expression in the RTN had no effect on chemosensitivity of breathing. $CO_2$-dependent ATP release (most probably via Cx26) has been observed in vivo from the rostral ventral surface of the medulla, but this is more medial and probably closer to the Raphé magnus rather than the RTN, which is substantially more lateral[26,34]. Our data would suggest that chemosensory mechanisms within the RTN are completely independent of Cx26 and may not involve the direct sensing of $CO_2$. The observation of pH-dependent ATP release from RTN astrocytes[21] via a membrane process involving the sodium bicarbonate transporter (NBCe1) and reversed $Na^+$–$Ca^{2+}$ exchange[59] supports this hypothesis.

There is compelling evidence to support the serotonergic medullary raphé neurons as an important mediator of respiratory chemosensitivity. Selective DREADD-mediating silencing of these neurons gave a substantial reduction (~40%) of the adaptive change in ventilation to a 5% inspired $CO_2$ challenge[15]. Recently, chemosensory responses in RTN neurons have been shown to depend partially on serotonergic inputs from raphé neurons and RTN neurons have been proposed as relays of chemosensory information rather than primary chemosensors[60]. Interestingly, the serotonergic raphé system extends caudally and will be medial to the chemosensory glial cells that we have identified in this study. As the surface glial cells that we transduced with $Cx26^{DN}$ have long processes that project both rostrally and medially, it is plausible that these cells could excite these raphé neurons via release of ATP or other gliotransmitters. Furthermore, there are additional pH-sensitive serotonergic neurons in the caudal parapyramidal area[48] that are likely to be intermingled with the glial cells that we describe in this study. $CO_2$-dependent ATP release from these glial cells could potentially excite the neighbouring serotonergic neurons. As these serotonergic neurons project widely to other chemosensory areas of the medulla including the raphé obscurus, raphé pallidus, and the nucleus tractus solitarius[49], their potential activation via release of ATP from $CO_2$-sensitive glia in the same region gives a further mechanism by which this chemosensory signal could converge with that mediated via other chemosensory neurons (including the pH-dependent chemosensory pathway) and be propagated within the brain stem neural networks to facilitate breathing.

Peripheral (mainly CB) and central chemoreceptors mediate the $CO_2$-dependent control of breathing. Overall, about 30% of the total chemosensory response is mediated peripherally and the remainder via central chemoreceptors[61]. In addition, there are two components to the adaptive changes in ventilation—an increase in respiratory frequency and an increase in tidal volume[1]. These two components combine to increase the rate of ventilation of the lungs: minute ventilation. Broadly, peripheral

chemoreceptors appear primarily to increase respiratory frequency, whereas activation of central chemoreceptors have a more powerful effect on tidal volume[3,62,63]. However, ATP release from astrocytes in the preBotzinger complex does modulate changes in respiratory frequency in response to elevated $CO_2$[64].

The widely accepted explanation for the chemosensitivity of breathing posits that pH-sensitive neurons detect changes in the pH of arterial blood and then excite the medullary networks that control breathing. It is clear that the peripheral chemoreceptors are exclusively sensitive to pH[65]. pH-sensitive TASK-1 channels contribute to the detection of pH changes in blood in the CB glomus cells and thus contribute to peripheral chemoreception[17]. Centrally, neurons from the RTN[8], the caudal parapyramidal area[48], and the medullary raphé[50,66] respond to changes in pH most probably via TASK-2[11,19,67–70] and potentially GPR4[11,20].

Given this multiplicity of evidence supporting the role of pH detection on regulation of breathing, why should $CO_2$ sensing via Cx26 be particularly important? First, the contribution of central direct $CO_2$ sensing to the regulation of breathing is about 50% of the centrally generated chemosensory response[23]. Viral transduction by $Cx26^{DN}$ reduced the mean adaptive ventilatory response to 6% inspired $CO_2$ by about 30%—mainly via a reduction of the increase in tidal volume. It is unlikely that we transduced all of the chemosensitive glial cells in this area, so this may be an underestimate of the contribution of this mechanism. Pharmacological blockade of Cx26 in the medulla, with imperfectly selective and potent agents, gave a reduction in the adaptive ventilatory response in anaesthetised and ventilated rats of about 20%[26]. The $Cx26^{DN}$ transduction is clearly more effective than the prior pharmacological approach—it is highly selective and will completely remove Cx26-mediated $CO_2$ sensitivity from the glial cells in which it is expressed.

As central chemoreceptors mediate about 70% of the adaptive response, $CO_2$ sensing via Cx26 in the caudal parapyramidal area mediates just under half of the total central ventilatory response to modest levels of hypercapnia. A further comparison to give physiological context is that this contribution from Cx26 is slightly smaller than, but broadly comparable to, the DREADD-mediated inactivation of the entire population of raphé serotonergic neurons, which gave a 40% reduction in the adaptive ventilatory response at similar levels of inspired $CO_2$[15]. The physiological importance of this contribution to chemosensing is further confirmed by the evolutionary conservation of the carbamylation motif for >400 million years and $CO_2$ sensitivity in Cx26 in all amniotes[29].

The contribution of Cx26 to respiratory chemosensitivity occurred over an intermediate (~6%), but physiologically important, range of inspired $CO_2$. There was no detectable effect of $Cx26^{DN}$ on the ventilatory response to 3% inspired $CO_2$. This is unlikely to reflect the properties of Cx26, as it is sensitive to changes in $PCO_2$ from 20 to 70 mmHg[27,33], so would very likely be capable of detecting the small change in systemic $PCO_2$ resulting from a 3% $CO_2$ challenge. Cx26 contributes only to the adaptive change in tidal volume during hypercapnia (Fig. 5). The ventilatory response to 3% $CO_2$ involved only a small change in tidal volume (an increase of 0.34 µl/g over $V_T$ at 0% $CO_2$ in $Cx26^{WT}$-injected mice, Fig. 5 and Supplementary Table 1). If Cx26 contributed about 30% of this increase, the expected difference in the $V_T$ between $Cx26^{WT}$- and $Cx26^{DN}$-injected mice would be ~0.1 µl/g. Although there is a trend toward a reduced change in $V_T$ at 3% $CO_2$ (Fig. 5 and Supplementary Table 1), our experiments were insufficiently powered to detect such a small difference and hence cannot reveal any possible contribution of Cx26 at this level of inspired $CO_2$. A further factor in determining the overall contribution of cells that express Cx26 to respiratory

control will be the way they and other populations of chemosensory cells, both central and peripheral, connect to the neuronal circuits controlling breathing.

At 9% inspired $CO_2$, Cx26-mediated chemosensing makes no contribution to the regulation of breathing, and the ventilatory responses to these higher levels are most likely exclusively mediated by changes in pH. This may be because the acidification caused by the higher level of inspired $CO_2$ can counteract the $CO_2$-dependent opening of Cx26 hemichannels. Cx26 hemichannels are closed by strong acidification[31]. We have previously reported that acidification reduces the conductance change evoked by a $PCO_2$ stimulus in Cx26-expressing HeLa cells[27]. Furthermore, the same increase in $PCO_2$ at pH 7.35 evokes about half the ATP release from the chemosensory cells in the ventral medulla compared to that evoked at pH 7.5[26]. This is presumably because acidification makes carbamylation of the critical lysine residue harder to achieve[28].

The caudal parapyramidal area (Fig. 7) was the only location in which we found a contribution of Cx26 contributed to respiratory chemosensitivity. This area has been previously described as containing pH-sensitive serotonergic neurons[48]. The caudal parapyramidal area is thus sensitive to both chemosensory stimuli and is a potential point of convergence of the $CO_2$-mediated and pH-mediated chemosensory signals. In the rostral medulla, our injections of Cx26[DN] were too lateral to test whether Cx26 expression in more medial glial cells might also contribute to activation of the pH-sensitive raphé magnus neurons, but this would be an interesting hypothesis to investigate.

To conclude, the development of Cx26[DN] as a tool to remove $CO_2$ sensitivity from endogenously expressed Cx26 has provided the first genetic evidence, to our knowledge, for the involvement of Cx26 in the control of breathing. Our data provide a mechanistic link between the binding of $CO_2$ to a structural motif on Cx26 and shows that direct sensing of $CO_2$ by glial cells in a circumscribed area of the ventral medulla contribute nearly half of the centrally generated adaptive ventilatory response to $CO_2$.

## Methods

**Animals**. All animal procedures were evaluated by the Animal Welfare and Ethical Review Board of the University of Warwick and carried out in strict accordance with the Animals (1986) Scientific Procedures Act of the UK under the authority of Licence PC07DE9A3. Mice were randomly assigned to their groups (Cx26[WT], Cx26[DN], and sham). A total of 80 mice were used in this study.

*Pilot experiment, and RTN*: Male and female mice aged 12–20 weeks were used. The mice had a floxed Cx26 allele on a C57BL6 background (EMMA strain 00245). These mice were not crossed with any cre lines so had normal WT expression of Cx26.

*Main caudal parapyramidal experiment*: C57BL6 WT male mice aged 12–17 weeks were used.

*Very caudal experiment*: C57BL6 WT Male mice aged 12–20 weeks were used.

**Cell lines**. HeLa DH (obtained from ECACC) and stable Cx26-expressing HeLa cells (gift from K. Willecke) were maintained in Dulbecco's modified Eagle's medium supplemented with 10% foetal calf serum, 50 µg/mL penicillin–streptomycin, and 3 mM $CaCl_2$ at 37 °C.

**Construction of connexin gene constructs**. *Connexin43* DNA sequence from *Rattus norvegicus* was purchased from Addgene (mCherry-Cx43-7, plasmid #55023, gifted by Michael Davidson) and subcloned into a PUC 19 vector such that the transcript would form a fusion protein with whichever fluorophore we engineered to be 3′ (downstream) of Cx43 (as with Cx26 constructs).

Dominant-negative mutant Cx26 (Cx26[DN]) DNA with R104A K125R mutations was produced through two steps. First, Cx26 DNA with the K125R mutation and omitted STOP codon (to allow for fusion proteins) was synthesised by Genscript USA from the Cx26 sequence (accession number NM_001004099.1) and subsequently subcloned into a PUC 19 vector such that the transcript would form a fusion protein with mCherry at the C-terminus of Cx26. Second, the R104A mutation was introduced by Agilent Quikchange site-directed mutagenesis, using PUC19-Cx26(K125R) DNA as the template. Primer sequences for the mutagenesis were as follows: Cx26(R104A) forward 5′ GGC CTA CCG GAG ACA CGA AAA

GAA AGC GAA GTT CAT GAA GG 3′, Cx26(R104A) reverse 5′ CCT TCA TGA ACT TCG CTT TCT TTT CGT GTC TCC GGT AGG CC 3′.

Design and characterisation of the Clover and mRuby2 protein fluorophores used in these experiments was published by ref. [36]. mRuby2 DNA was purchased from Addgene (mRuby2-C1, plasmid #54768, gifted by Michael Davidson) and subcloned into a PUC 19 vector so that it was 3′ of whichever connexin we engineered to be 5′ (upstream) of mRuby2. Clover DNA was a gift from Sergey Kasparov, Bristol and was subcloned into a PUC 19 vector so that it was 3′ of whichever connexin we engineered to be 5′ (upstream) of Clover.

**HeLa cell transfection**. To express the desired constructs, HeLa cells were transiently transfected with 0.5 µg of DNA of each pCAG–connexin–fluorophore construct to be co-expressed (1 µg total), using the GeneJuice transfection agent protocol (Merck Millipore).

**Experimental artificial CSF (aCSF) solutions**. *Control (35 mmHg $PCO_2$)*: 124 mM NaCl, 26 mM NaHCO₃, 1.25 mM NaH₂PO₄, 3 mM KCl, 10 mM D-glucose, 1 mM MgSO₄, 2 mM CaCl₂. This was bubbled with 95%O₂/5% CO₂ and had a final pH of ~7.4.

*Hypercapnic (55 mmHg $PCO_2$)*: 100 mM NaCl, 50 mM NaHCO₃, 1.25 mM NaH₂PO₄, 3 mM KCl, 10 mM D-glucose, 1 mM MgSO₄, 2 mM CaCl₂. This was bubbled with sufficient CO₂ (approximately 9%, balance O₂) to give a final pH of ~7.4.

*Zero $Ca^{2+}$*: 124 mM NaCl, 26 mM NaHCO₃, 1.25 mM NaH₂PO₄, 3 mM KCl, 10 mM D-glucose, 1 mM MgSO₄, 2 mM MgCl₂, 1 mM EGTA. This was bubbled with 95%O₂/5% CO₂ and had a final pH of ~7.4.

**Dye-loading assay**. We used a dye-loading protocol that has been developed and extensively described in our prior work[27–29,71,72]. HeLa-DH cells were plated onto coverslips and transfected with Cx26[DN]. Dye-loading experiments were performed 24–72 h after transfection. For experiments involving co-expression of Cx26[DN] and Cx26[WT], HeLa cells that stably expressed Cx26[WT] were transfected with Cx26[DN]. Dye loading was performed over a 3-day period, beginning at 4 days post-transfection.

To perform the dye loading, cells washed in control aCSF were then exposed to either control or hypercapnic solution containing 200 µM 5(6)-carboxyfluorescein (CBF) for 10 min. Subsequently, cells were returned to control solution with 200 µM CBF for 5 min, before being washed in control solution without CBF for 30–40 min to remove excess extracellular dye. A replacement coverslip of HeLa cells was used for each condition. For each coverslip, mCherry staining was imaged to verify Cx26 expression.

To quantify fluorescence intensity, cells were imaged by epifluorescence (Scientifica Slice Scope (Scientifica Ltd, Uckfield, UK), Cairn Research OptoLED illumination (Cairn Research Limited, Faversham, UK), ×60 water Olympus immersion objective, NA 1.0 (Scientifica), Hamamatsu ImageEM EMCCD camera (Hamamatsu Photonics K.K., Japan), and Metafluor software (Cairn Research). Using ImageJ, regions of interest (ROIs) were drawn around individual cells and the mean pixel intensity for each ROI was obtained. The mean pixel intensity of the background fluorescence was also measured in a representative ROI, and this value was subtracted from the measures obtained from the cells. This procedure was used to subtract the background fluorescence from every pixel of all of the images displayed in the figures. At least 40 cells were measured in each condition, and the mean pixel intensities were plotted as cumulative probability distributions.

The experiments were replicated independently (independent transfections) at least five times for each Cx26 variant and condition. All experiments performed at room temperature.

**FRET data capture and analysis**. For FRET analysis, HeLa cells co-transfected with Cx-Clover (donor) and Cx-mRuby2 (acceptor) 72 h after transfection were washed 3× with phosphate-buffered saline (PBS), fixed with 4% paraformaldehyde (PFA) for 20–30 min, washed a further 3× with PBS, and then stored in PBS at 4 °C. FRET studies were carried out within 2 weeks of fixation. They examined with a Zeiss LSM 710 Confocal microscope; C-Apochromat ×63/1.20 W Korr M27. Two channels were recorded: 495–545 nm (clover) and 650–700 nm (mRuby2), and images were acquired sequentially with 458- and 561-nm argon lasers, respectively. Imaging parameters for the clover channel are as follows: power, 30.0; pinhole, 78.5; gain (master), 700; digital offset, 0; digital gain, 1.0. Imaging parameters for the mRuby2 channel are as follows: power, 30.0; pinhole, 78.5; gain (master), 750; digital offset, 0; digital gain, 1.0. Photobleaching was performed using the 561-nm laser (as it only excites mRuby2) for 80 frames at 100% power, targeting ROIs. Image acquisition was as follows: ROIs were selected and drawn, including a background region and an ROI that was not to be bleached; pre-bleaching images were acquired for each channel; mRuby2 was photobleached; post-bleaching images were acquired for each channel. Acquisition parameters were kept identical across samples to allow comparison of results.

Pixel intensities from each background-adjusted ROI were used to calculate FRET efficiency (*E*), bleaching efficiency (*B*), relative donor abundance (*D*), relative acceptor abundance (*A*), relative donor–acceptor ratio (DA ratio), relative acceptor

quantity ($A$ level), and donor–acceptor distance ($R$) as follows:

$$E = \frac{Clover_{post} - Clover_{pre}}{Clover_{post}}, \tag{1}$$

$$D = \frac{Clover_{post}}{Clover_{post} + mRuby2_{pre}}, \tag{2}$$

$$A = \frac{mRuby2_{pre}}{Clover_{post} + mRuby2_{pre}}, \tag{3}$$

$$DA\ ratio = \frac{Clover_{post}}{mRuby2_{pre}}, \tag{4}$$

$$A\ level = mRuby2_{pre}. \tag{5}$$

Colocalisation threshold calculations were carried out in ImageJ using the Costes method[73]: 100 iterations, omitting zero–zero pixels in threshold calculation.

**Lentivirus (LV) design and production.** Two LV constructs were designed to introduce either the Cx26[DN] or Cx26[WT] gene into the host cell genome (Supplementary Fig. 3). Constructs are ~4900 bp in length. The sequence of interest consisted off the Cx26 gene variant (Cx26[WT] or Cx26[DN]) immediately followed by an IRES and clover sequence (Cx26:IRES:Clover). The IRES sequence was from the encephalomyocarditis virus[74]. LV constructs were produced and packaged by Cyagen Biosciences (USA) using the third-generation packing system. Constructs had a titre >10[8] TU/mL, as confirmed by quantitative PCR on genomic DNA extracted from the infected cells.

**Stereotaxic viral transduction.** Anaesthesia was induced by inhalation of isoflurane (4%). The mouse was then placed on a thermocoupled heating pad (TCAT-2LV, Physitemp) to maintain body temperature at 33 °C and headfixed into a stereotaxic frame. A face mask was used to maintain anaesthesia (isoflurane, intranasal, 0.5–2.5%). Atropine was provided (subcutaneous, 0.05 mg/kg) before surgery to stop pleural effusion. Adequacy of anaesthesia was assessed by respiratory rate, body temperature, and pedal withdrawal reflex. Preoperative Meloxicam (subcutaneous, 2 mg/kg) and postoperative Buprenorphine (intraperitoneal, 0.05 mg/kg) were provided for analgesia. If any animal showed signs of pain in the days following surgery, additional analgesia was administered as required. No animals displaying signs of pain or receiving analgesia were used in plethysmographic recordings.

To maintain consistent placement of the injection pipette, the intra-aural line was adjusted so that bregma was level to a point on the skull 2 mm caudal to bregma. Two small holes were made in the interparietal plate to allow for bilateral injection of virus particles via a micropipette lowered into the correct position via a stereotaxic manipulator. The injections were performed manually, using a 1-ml syringe, at a rate ~200 nl/min. A total of 350–400 nl of undiluted virus or saline was injected per side of the brain; the experimenter was blind to the injection solution. Co-ordinates (mm) relative to bregma were: RTN: 5.7 and 5.9 caudal (two injections per side of brain), 1.1 lateral and 5.3–5.9 ventral, injection arm at 9° to vertical; caudal parapyramidal area: 5.95 caudal, 0.8 lateral, and 5.7–6.1 ventral, injection arm at 0° to vertical; very caudal area: 6.2 caudal, 0.8 lateral and 5.7–6.2 ventral, injection arm at 0° to vertical; pilot experiment: 5.9 caudal, 1.1 lateral, and 5.5–6.0 ventral, injection arm at 0° to vertical. Co-ordinates were confirmed by post hoc immunostaining for viral-driven expression of fluorophores. Data were only included from mice whose injection sites were within the correct location.

**Immunohistochemistry.** Mice were culled by overdose of isoflurane or intraperitoneal injection of sodium pentobarbital (>100 mg/kg) and transcardially perfused with PFA (4%). The brain was harvested and post-fixed in 4% PFA for 24 h at 4 °C (to further increase tissue fixing), before being transferred to 30% sucrose (for cryoprotection) until the brain sunk to the bottom of the sucrose—usually ~2–3 days. PFA and sucrose were made in PBS.

For sectioning, brainstems were isolated, mounted with Tissue-Tek optimum cutting temperature compound (Sakura Finetek), and cut either sagittally or coronally at 40 μm on a cryostat (Bright Instruments). Sections were arranged in order, in 24-well plates, and stored in PBS. For immunostaining, free-floating sections were incubated at room temperature overnight in PBS containing 0.1% Triton X-100 (Sigma-Aldrich, UK) (0.1% PBS-T) and the appropriate primary antibodies: goat anti-choline acetyltransferase (1:100) (Merck UK, ab144), chicken anti-GFAP (1:500) (abcam, ab4674), and rabbit anti-GFAP (1:500) (abcam, ab68428). Sections were then washed in PBS for 6 × 5 min before incubation at room temperature for 2.5–4 h in 0.1% PBS-T containing the appropriate secondary antibodies: donkey anti-goat Alexa Fluor 594 (1:250) (abcam, ab150132), goat anti-rabbit Alexa Fluor 594 (1:250) (abcam, ab150080), and goat anti-chicken Alexa Fluor 488 (1:250) (abcam, ab150169). After secondary antibody incubation, sections were again washed in PBS for 6 × 5 min before mounting onto polylysine-coated slides (Polysine, VWR). Mounted slices were left to dehydrate overnight

before applying coverslips using either Aqua-Poly/Mount (Polysciences Inc., Germany) or Fluoroshield with 4,6-diamidino-2-phenylindole (Sigma-Aldrich, UK). All steps were performed at 21 °C, and during any period of incubation or washing, sections were gently agitated on a shaker.

**Whole-body plethysmography.** A custom built plethysmograph, constructed from a plexiglass box (0.5 l), was equipped with an air-tight lid, a pressure transducer to detect the breathing movements of the mouse, and gas inlets and outlets to permit gas flow through the chamber. A heated (via a water bath) mixture of $O_2$ (~20%), $N_2$ (80%), and $CO_2$ (0–9%) flowed through the chamber regulated to a rate of 1 l/min. The amount of $O_2$ and $CO_2$ in the mixture was measured, just prior to entry into the chamber, by a gas analyser (Hitech Instruments, GIR250 Dual Sensor Gas Analyser). Pressure signals were recorded with an NL108T2 – Disposable Physiological Pressure Transducer (Digitimer) amplified and filtered using the NeuroLog system connected to a 1401 interface and acquired on a computer using the Spike2 software (Cambridge Electronic Design). Airflow measurements were used to calculate: tidal volume ($V_T$: signal trough at the end of expiration subtracted from the peak signal during inspiration, converted to ml following calibration with a 1-ml syringe), respiratory frequency ($f_R$: breaths per minute), and minute ventilation ($V_E$) (calculated as $V_T \times f_R$). The temperature inside the plethysmograph was maintained at ~31 °C, thermoneutral for C57BL/6 mice. The experimenter was only unblinded to the identity of the injected virus once acquisition and analysis of all plethysmographic recordings had been performed.

**Statistics and reproducibility.** For the dye-loading experiments, box–whisker plots were made and statistical tests were performed on the median change in pixel intensity where each replicate was a different transfection of the cells. The statistical package R was used for this analysis. For the FRET data, each replicate was a single ROI, and linear regression analysis and plots were carried out in GraphPad Prism. The plethysmographic data, each replicate was a single mouse, was analysed in SPSS via a two-way mixed-effects, repeated-measures ANOVA. Multiple post hoc comparisons were checked by the false discovery rate method, with the maximum allowed false discovery rate set to 0.05[75]. All reported comparisons pass this test. All post hoc comparisons were one sided as there was a clear a priori hypothesis that Cx26[DN] should decrease $CO_2$ sensitivity. Power calculations for the experiment in Fig. 5 were based on the pilot data of Supplementary Fig. 5 and were performed in GPower 3.1 for a two-way ANOVA with repeated measures, within- and between-factor interactions. Further statistical details can be found in the relevant figure legends.

**Reporting summary.** Further information on research design is available in the Nature Research Reporting Summary linked to this article.

## Data availability
All data generated or analysed during this study are included in this published article. The source data used to generate the charts in the paper are provided in Supplementary Data 1–3.

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

## Acknowledgements
We thank the MRC (MR/N003918/1) for support. J.V.d.W. was a doctoral student supported by the MRC (MR/J003964/1). N.D. was a Royal Society Wolfson Research Merit Award Holder.

## Author contributions
Molecular biology: J.V.d.W., L.M., J.C.; in vitro experiments: L.M., J.V.d.W., S.N.; immunocytochemical staining: J.V.d.W., A.B.; surgical procedures: J.V.d.W., A.B., R.H.; plethysmography: J.V.d.W.; data analysis: J.V.d.W., L.M., S.N., N.D.; study design: N.D., J.V.d.W., R.H.; writing paper: N.D., J.V.d.W., R.H., A.B.; all authors commented on drafts of the paper.

## Competing interests
The authors declare no competing interests.
