## [Peer Review File · Communications Biology]

Reviewers' comments:

Reviewer #1 (Remarks to the Author):

Understanding cellular and molecular transduction mechanisms for chemosensory control of breathing by CO₂ in the mammalian nervous system is an important biological problem of widespread interest in neuroscience and physiology. Most studies addressing this issue have focused on CO₂ detection via the associated changes in pH due to the altered balance between CO₂ and HCO₃ during hypercapnia in specific regions of the mammalian brainstem. This paper provides some new evidence for direct CO₂ sensing (i.e., not mediated by secondary changes in pH), involving glial (astrocyte) connexin 26 hemichannels (Cx26) in a circumscribed region (caudal parapyramidal area) of the mouse ventral medulla that has previously been implicated in chemosensory control of breathing in rodents (rat). This important finding indicates that CO₂-binding to Cx26 of astrocytes can be a transduction mechanism contributing to chemosensory regulation of breathing in concert with pH-sensing mechanisms operating in other specific medullary regions that have been extensively studied, particularly the retrotrapezoid nucleus (RTN). Also, of considerable interest is the authors' data indicating that chemosensing by RTN, where there is previous evidence for astrocyte involvement, does not appear from the present data to involve CO₂-sensing by Cx26. Also, the authors present interesting observations that astrocytes with unusual morphology at the ventral surface of the medulla in the caudal parapyramidal area may be involved.

From a technical standpoint, the molecular evidence presented is reasonably convincing and supports the authors' hypothesis that the specific mutations of Cx26 employed to interfere with carbamylation and carbamate bridges produces a dominant negative subunit (Cx26DN) with an effect on CO₂ sensitivity of Cx26WT, due to Cx26DN co-assembly into heteromeric hexamers with Cx26WT, and eliminating CO₂ binding capabilities. The authors developed novel viral constructs to target astrocytes and transduce Cx26DN in vivo in a site-specific way in the mouse medulla. This is a very nice experimental approach that allowed identification of Cx26 function in the caudal parapyramidal area. In general, the authors have developed a new tool to remove CO₂ sensitivity from endogenously expressed Cx26 in vivo and produced evidence for the involvement of glial Cx26 in chemosensory control of breathing.

The experiments are technically sound, the data analyses including statistical analyses are appropriate, the results are important for the field, and the manuscript is generally well written. The supplementary material presented is useful and strengthens the paper. I have very few suggestions for revision.

Suggestions for revision.

Abstract

The abstract text should more explicitly indicate that the blunting of changes in tidal volume and minute ventilation in mice occurred only at the moderate levels of hypercapnia (6% inspired CO₂ challenge).

Supplementary Figures

Figure S4 legend. The authors should note that the large amplitude excursions in the raw plethysmography data show in panel (a) represent movement artefacts (?).

Reviewer #2 (Remarks to the Author):

Van de Wiel et al. provide novel evidence indicating that the expression of a dominant negative subunit (Cx26DN), co-assembled into the Cx26 hexamer, reduced the CO₂-sensitivity from

endogenously expressed wild type Cx26 assessed in vitro. Interestingly, the selective expression of this Cx26DN in astrocytes of the caudal parapyramidal area of mice blunted the respiratory response induced by hypercapnia.

The results are relevant for the understanding of the mechanisms underlying central chemoreception, a key property distributed along several brainstem nuclei, that allows to match breathing to the physiological demands. In addition, these results provide new evidence supporting the role of astrocytes in central chemoreception.

Overall, the manuscript is clearly written and exacting, the methodology is appropriate and pertinent, the experiments were well-conducted, and the results obtained were straightforward. However, there are some points that require clarifications:

General comments

1) Two results are puzzling. First, the narrow range of FiCO₂ (around 6%) in which the ventilatory responses are reduced after bilateral viral transfection to express Cx26DN in astrocytes (Fig. 5). The authors discuss that, during inhalation of 9% CO₂, Cx26 hemichannels are closed by acidification and that acidification also makes carbamylation of the critical lysine residue in Cx26 harder to achieve. This explanation implies that another mechanism should be recruited or enhanced to compensate totally the reduction in the Cx26 contribution to central chemoreception. This is a reasonable explanation for extreme hypercapnia. The absence of effect during inhalation of 3% CO₂, implies that Cx26 mechanism is not relevant or redundant during 3% CO₂ inhalation. Is there any property of the CO₂-binding motif to Cx26 that could be associated to this stimulus range constrain? Second, the overexpression of Cx26WT does not enhance CO₂-sensitivity and ventilatory responses to hypercapnia (supplementary Fig. 7). This absence of effects in presence of Cx25WT overexpression deserves a brief discussion.

2) Since Cx26 can participate as gap-junction component, and gap-junctions endow astrocytes with important functions, authors may consider a brief discussion to analyze a possible impairment in gap-junction formation or regulation as explanation to their results.

3) Hemichannel opening is influenced by Ca²⁺ concentration and voltage. It seems clear in the experiments illustrated in Figure 4, that hemichannel CO₂-sensitivity is abolished when Cx26DN is expressed in Cx26-HeLa cells, whereas zero Ca²⁺ condition reveals that other hemichannel functionality is preserved. It is reported that in syndromic Cx26 mutations, the dysfunctions appear as result from dysregulation of Cx26 undocked hemichannels by voltage or extracellular calcium. Only for a more detailed description of the Cx26DN construct, do the authors have some idea about the effects of this dominant negative subunit in the voltage-dependence of Cx26?

Specific comments

1) In regard to the point 2 (above), the authors should provide the basal values for VT and VE, in addition to the normalized curves in Figure 5 and supplementary Fig.7. It is not clear what represents the arbitrary units they used. Are they expressed as ratio of any basal value? Are the VT and VE normalized by the weight of the mice?

2) In Figure 3, ANOVA P-value is not indicated for "a" and "b" figures. Please, also indicate in the legend which post hoc test was used. Significant and non-significant differences between groups should be indicated in figures "a" and "b".

3) In Figure 5, P values of two-way mixed ANOVA are not indicated in the comparisons with significant statistical differences. Please, indicate which post hoc test was used to obtain the P values showed for VT and VE at week 2.

Reviewer #3 (Remarks to the Author):

The study "Connexin26 mediates CO₂-dependent regulation of breathing via glial cells of the medulla oblongata" by Van de Wiel et al, aimed to develop a dominant negative Cx26 subunit (Cx26DN) to remove its CO₂-sensitivity in order to test the involvement of Cx26 in the breathing response to hypercapnia. They found that expression of Cx26DN in glial cells of the caudal

parapyramidal area reduced the change in breathing in response to 6% Co₂. Although, the authors claimed that this reduction was of approximately 30% it is hard to find this exact change in the figures. This is a very well designed and conducted study that reveal a very specific cellular and molecular mechanism involved in the hypercapnic modulation of breathing. The only two major concerns I have is that despite that a likely explanation is offered in the Discussion, it is puzzling that the effect of Cx26 manipulation on breathing is only observed at 6% Co₂. It is indicated that the acidosis induced at higher Co₂ concentration abolishes the participation of Cx26. However, I would love to see if this possibility is evaluated in the in vitro system used in this study. Moreover, it would be worth to Discuss the relevance of this cellular mechanism (glial CX26) that might work only in a restricted range of Co₂ concentrations. Other point that is not assessed or discussed in the study is if the participation of Cx26 in CO₂ sensing involves its gap-junction and/or pannexin action? Is it possible to test these mechanisms? Next, I will enumerate a variety of issues that could help to improve the manuscript:

-For the Introduction and Discussion, it would be relevant to mention if Cx26 expression has been previously characterized in the area of interest.

-I would like to see, in the results section, a more detailed description of the data. For instance, it would be helpful to see central tendency and variability values, the N and the p of the statistical comparisons.

-Since most of the data presented in Fig. 1 is already published, I consider that such figure can be removed.

-It is mentioned that FRET interaction between Cx43 and 26 was "much lower than that of Cx26WTCx26WT or Cx26WT-Cx26DN", but the Figure 2 shows very similar FRET in all Cx combinations. Thus, the images in such Fig. might not be very representative.

-As a matter of interpretation, is it possible to speculate about the stoichiometry of the putative heteromers of Cx26WT-Cx26DN?

-Is it possible to correlate the amount of CO₂-induced dye-loading, at the different days tested, with the amount of Cx26DN expressed -by the attached fluorescent protein intensity-? Is there a dramatic change in Cx26DN expression between days 5 and 6? Or of its presence in the cell-membrane?

-For the area of interest, a quantification of the amount of GFAP and Cx26DN co-localized is required. A quantification of the amount of Cx26DN no co-localizing with GFAP is also required. Finally, a quantification of the amount of GFAP no co-localizing with Cx26DN is required too.

-A clear definition of the infected area in all animals is required, perhaps as an scheme, to have an idea of the area in which Cx26 is influencing the response to CO₂. This scheme can be built by the data collected from the corroborations obtained as follows "Co-ordinates were confirmed by post hoc immunostaining for viral driven expression of fluorophores. Data were only included from mice whose injection sites were within the correct location"

-An indication of the total amount of animals used for this study would be useful.

-Explain the conditions in which "additional analgesia was administered"

-A more detailed description of the "custom built plethysmograph" is required. For instance, the dimensions and characteristics of the plexiglass box. The characteristics and/or model of the pressure transducer.

-There are some sentences that require proper support by original references:

--"Breathing is a vital function that maintains the partial pressures of O₂ and CO₂ in arterial blood within the physiological limits. Chemosensory reflexes regulate the frequency and depth of breathing to ensure homeostatic control of blood gases"

--"pH-sensitive K⁺ channels (TASKs and KIRs) are potential transducers"

--"a subregion that has been more recently studied and termed the "caudal parapyramidal" area."

--"Peripheral (mainly carotid body) and central chemoreceptors mediate the CO₂-dependent control of breathing"

--"there are two components to the adaptive changes in ventilation - an increase in respiratory frequency, and an increase in tidal volume. These two components combine to increase the rate of ventilation of the lungs: minute ventilation"

-There are some typos that should be corrected

--“(Huckstepp et al., 2010b))”

--“(Cairn Research))”

--“Physiotemp”

-Something is missing in the following sentence:

--“Design and characterization of the and mRuby2 protein fluorophores”

-I did not understand the message of the following sentence:

--“The nose bar was adjusted so that Bregma was level to a point 2 mm caudal it”

Reviewer #4 (Remarks to the Author):

In this manuscript, Van de Weil et al. have investigated the role of Connexin26 (Cx26) in central carbon dioxide sensing. The authors provide compelling evidence that Cx26 is important in determining the central response to CO₂ changes by generating and expressing a dominant negative form of the hemichannel both in cultured cells and in an in vivo model of hypercapnia induced respiration. Interestingly, the authors propose a pH-independent role for CO₂ sensing via direct carbamylation of Cx26. This work is supportive of previous work from this group identifying a role for Cx26 in CO₂ sensing. This is a clearly written paper which contains important and compelling data in an area of significant biological interest (mechanisms of CO₂ sensing in mammals).

I have just two comments / suggestions that the authors may wish to consider.

1) In the introduction, the authors should discuss what (if anything) is known about regulation of Cx26 by factors other than CO₂. Do CO₂ levels alter the response of Cx26 to other stimuli. This possibility should at least be discussed.

2) Can the authors elaborate on whether they believe the carbamylation of Cx26 to be spontaneous or controlled. Is there evidence for an enzymatic process? Have the authors been able to demonstrate by, for example mass spectrometry that the double negative mutant is not carbamylated?

We thank the editors and reviewers for their careful reading of the MS and thoughtful comments. These have been very helpful in improving the paper.

To summarize, the referees agree in that his work will contribute to the field of chemosensory control of breathing, however, they raise concerns which we think should be addressed in the revised manuscript. Reviewer #1, #2, and #3 all raise concerns about the narrow range (6% CO₂) at which Cx26 exerts the proposed function, and question the biological implications of these findings. We believe this is an important issue that should be addressed in the revised manuscript.

We have given these issues further thought and considerably revised the discussion on pp 12 and 13 to take them into account.

Reviewer #2 and #3 also request additional discussion about the role of gap junctions in this process, and any potential effects on gap junction.

We have addressed this below and in the revised MS (pp 9, 10). We also provide additional data to demonstrate the Cx26^{DN} forms functional gap junctions, which are not sensitive to CO₂ (Supplementary Fig 10).

Reviewer #3 also request additional experiments and more transparency when reporting statistic and n number.

We have addressed the reviewer's points with reference to the published literature which shows the requested evidence (pp 12, 13). We have attempted to be as transparent in our reporting as possible by reporting every data point. We report the total animal number (pp 13). We have made the post hoc comparisons clearer (pp 6, 19), which we agree was not well explained.

Lastly, reviewer #4, the connexin expert, requests additional discussion about other factors affecting Cx26 function and empirical evidence to support the Cx26 carbamylation claim.

We have discussed these points with reference to the existing literature (pp 7, 8).

We have made a further change that has not been requested by the reviewers but has been stimulated by discussion of our preprint on bioRxiv. Consequently, we have added a paragraph to the Discussion pp 8, 9 to address the issue of the timing of the effects of Cx26^{DN}, known examples of compensation within the chemosensory networks, and possible mechanisms by which the effect of Cx26^{DN} could be compensated for.

Reviewers' comments:

Reviewer #1 (Remarks to the Author):

Understanding cellular and molecular transduction mechanisms for chemosensory control of breathing by CO₂ in the mammalian nervous system is an important biological problem of widespread interest in neuroscience and physiology. Most studies addressing this issue have focused on CO₂ detection via the associated changes in pH due to the altered balance between CO₂ and HCO₃ during hypercapnia in specific regions of the mammalian brainstem. This paper provides some new evidence for direct CO₂ sensing (i.e., not mediated by secondary changes in pH), involving glial (astrocyte) connexin 26 hemichannels (Cx26) in a circumscribed region (caudal parapyramidal area) of the mouse ventral medulla that has previously been implicated in chemosensory control of breathing in rodents (rat). This important finding indicates that CO₂-binding to Cx26 of astrocytes can be a transduction mechanism contributing to chemosensory regulation of breathing in concert with pH-sensing mechanisms operating in other specific medullary regions that have been extensively studied, particularly the retrotrapezoid nucleus (RTN). Also, of considerable interest is the authors' data indicating that chemosensing by RTN, where there is previous evidence for astrocyte involvement, does not appear from the present data to involve CO₂-sensing by Cx26. Also, the authors present interesting observations that astrocytes with unusual morphology at the ventral surface of the medulla in the caudal parapyramidal area may be involved.

From a technical standpoint, the molecular evidence presented is reasonably convincing and supports the authors' hypothesis that the specific mutations of Cx26 employed to interfere with carbamylation and carbamate bridges produces a dominant negative subunit (Cx26DN) with an effect on CO₂ sensitivity of Cx26WT, due to Cx26DN co-assembly into heteromeric hexamers with Cx26WT, and eliminating CO₂ binding capabilities. The authors developed novel viral constructs to target astrocytes and transduce Cx26DN in vivo in a site-specific way in the mouse medulla. This is a very nice experimental approach that allowed identification of Cx26 function in the caudal parapyramidal area. In general, the authors have developed a new tool to remove CO₂ sensitivity from endogenously expressed Cx26 in vivo and produced evidence for the involvement of glial Cx26 in chemosensory control of breathing.

The experiments are technically sound, the data analyses including statistical analyses are appropriate, the results are important for the field, and the manuscript is generally well written. The supplementary material presented is useful and strengthens the paper. I have very few suggestions for revision.

Suggestions for revision.

Abstract

The abstract text should more explicitly indicate that the blunting of changes in tidal volume and minute ventilation in mice occurred only at the moderate levels of hypercapnia (6% inspired CO₂ challenge).

Done

Supplementary Figures

Figure S4 legend. The authors should note that the large amplitude excursions in the raw plethysmography data show in panel (a) represent movement artefacts (?).

Done

Reviewer #2 (Remarks to the Author):

Van de Wiel et al. provide novel evidence indicating that the expression of a dominant negative subunit (Cx26DN), co-assembled into the Cx26 hexamer, reduced the CO₂-sensitivity from endogenously expressed wild type Cx26 assessed in vitro. Interestingly, the selective expression of this Cx26DN in astrocytes of the caudal parapyramidal area of mice blunted the respiratory response induced by hypercapnia.

The results are relevant for the understanding of the mechanisms underlying central chemoreception, a key property distributed along several brainstem nuclei, that allows to match breathing to the physiological demands. In addition, these results provide new evidence supporting the role of astrocytes in central chemoreception.

Overall, the manuscript is clearly written and exacting, the methodology is appropriate and pertinent, the experiments were well-conducted, and the results obtained were straightforward. However, there are some points that require clarifications:

General comments

1) Two results are puzzling. First, the narrow range of FiCO₂ (around 6%) in which the ventilatory responses are reduced after bilateral viral transfection to express Cx26DN in astrocytes (Fig. 5). The authors discuss that, during inhalation of 9% CO₂, Cx26 hemichannels are closed by acidification and that acidification also makes carbamylation of the critical lysine residue in Cx26 harder to achieve. This explanation implies that another mechanism should be recruited or enhanced to compensate totally the reduction in the Cx26 contribution to central chemoreception. This is a reasonable explanation for extreme hypercapnia. The absence of effect during inhalation of 3% CO₂, implies that Cx26 mechanism is not relevant or redundant during 3% CO₂ inhalation. Is there any property of the CO₂-binding motif to Cx26 that could be associated to this stimulus range constrain?

There are a number of interesting points here. Firstly, we don't think this reflects a property of CO₂ binding to Cx26 -from our prior work Cx26 is steeply sensitive to changes in PCO₂ from about 20 to 70 mmHg.

Secondly, Cx26 contributes to the change in tidal volume (V_T) that underlies the adaptive ventilatory response and not the change in respiratory frequency (f). The response to 3% CO₂ is mainly a change in f with very little change in V_T : a mean increase of 0.34 μ l/g was seen going from 0% to 3% in WT, of which Cx26 could provide about 30%; this translates to a potential decrease in the response of \sim 0.1 μ l/g with Cx26^{DN}. While there are hints of such a change in our data, our experiments were not sufficiently powered (statistically) to detect such a small change. On reflection, we think it is more likely that we cannot detect any contribution of Cx26 rather than we have shown that such a contribution does not exist. Parenthetically, we point out that an impractical number of mice would be needed to demonstrate (or rule out) a contribution of Cx26 to the response to 3% CO₂.

Finally, a further mechanistic factor is how different chemosensory areas are wired up to the breathing rhythm generator. This is likely to determine their relative contributions. This gives another reason why we see no effect.

We thank the reviewer for raising this issue as we had not properly discussed our results at 3% CO₂, and have now addressed these points in detail on pp 12.

Second, the overexpression of Cx26WT does not enhance CO₂-sensitivity and ventilatory responses to hypercapnia (supplementary Fig. 7). This absence of effects in presence of Cx25WT overexpression deserves a brief discussion.

We have added discussion of this point on pp 9.

2) Since Cx26 can participate as gap-junction component, and gap-junctions endow astrocytes with important functions, authors may consider a brief discussion to analyze a possible impairment in gap-junction formation or regulation as explanation to their results.

FRET shows that Cx26^{DN} still forms gap junctions. Cx26^{DN} abolishes GJ sensitivity to CO₂ but otherwise the gap junctions are functional. We discuss this point on pp9. We have added data to the supplementary information to document this point (Supplementary Figure 10).

3) Hemichannel opening is influenced by Ca²⁺ concentration and voltage. It seems clear in the experiments illustrated in Figure 4, that hemichannel CO₂-sensitivity is abolished when Cx26DN is expressed in Cx26-HeLa cells, whereas zero Ca²⁺ condition reveals that other hemichannel functionality is preserved. It is reported that in syndromic Cx26 mutations, the dysfunctions appear as result from dysregulation of Cx26 undocked hemichannels by voltage or extracellular calcium. Only for a more detailed description of the Cx26DN construct, do the authors have some idea about the effects of this dominant negative subunit in the voltage-dependence of Cx26?

There is no voltage dependence of Cx26 over relevant voltage ranges to glial cell function. We give some more information on this in the introduction, pp4.

Specific comments

1) In regard to the point 2 (above), the authors should provide the basal values for VT and VE, in addition to the normalized curves in Figure 5 and supplementary Fig.7. It is not clear what represents the arbitrary units they used. Are they expressed as ratio of any basal value? Are the VT and VE normalized by the weight of the mice?

We apologize for not making this clear in the MS. VT is normalized to body weight. Basal values at for VT and VE are provided (at 0%). The plethysmograph was calibrated for every run, and therefore the measurements are actually in $\mu\text{l}\cdot\text{g}^{-1}$. When we sent a draft of the paper out for comment, we were advised by a trusted collaborator that there are confounding factors inherent to the specific plethysmography apparatus and that values are not necessarily comparable between different plethysmographs and to use instead the term arbitrary units. However, as this has given rise to confusion and as, with the exception of the pilot study which used a slightly larger recording chamber (reported in Supplementary Figure 5), we used the same plethysmograph for all measurements in the study, we now give the results in $\mu\text{l}\cdot\text{g}^{-1}$.

2) In Figure 3, ANOVA P-value is not indicated for "a" and "b" figures. Please, also indicate in the legend which post hoc test was used. Significant and non-significant differences between groups should be indicated in figures "a" and "b".

We now show the post hoc comparisons on the figure for panel a. We did not perform any statistical analysis on panel b as the effect is completely obvious -there is no overlap between the unbleached ROIs and the other ROIs.

3) In Figure 5, P values of two-way mixed ANOVA are not indicated in the comparisons with significant statistical differences. Please, indicate which post hoc test was used to obtain the P values showed for VT and VE at week 2.

This oversight has now been rectified in the text and on the figure where ANOVA and the post-hoc p values are given.

Reviewer #3 (Remarks to the Author):

The study "Connexin26 mediates CO₂-dependent regulation of breathing via glial cells of the medulla oblongata" by Van de Wiel et al, aimed to develop a dominant negative Cx26 subunit (Cx26DN) to remove its CO₂-sensitivity in order to test the involvement of Cx26 in the breathing response to hypercapnia. They found that expression of Cx26DN in glial cells of the caudal parapyramidal area reduced the change in breathing in response to 6% Co₂. Although, the authors claimed that this reduction was of approximately 30% it is hard to find this exact change in the figures. This is a very well designed and conducted study that reveal a very specific cellular and molecular mechanism involved in the hypercapnic modulation of breathing. The only two major concerns I have is that despite that a likely explanation is offered in the Discussion, it is puzzling that the effect of Cx26 manipulation on breathing is only observed at 6% Co₂. It is indicated that the acidosis induced at higher Co₂ concentration abolishes the participation of Cx26. However, I would love to see if this possibility is evaluated in the in vitro system used in this study. Moreover, it would be worth to Discuss the relevance of this cellular mechanism (glial CX26) that might work only in a restricted range of Co₂ concentrations.

We appreciate that working out the size of the effect of Cx26^{DN} involves inferring the change from the graphs, which is a bit tricky. To address this, we have added two tables to the Supplementary Information to make this information more accessible.

We have added further discussion on the sensitivity of Cx26 and the effect on breathing to pp 12, 13. We note that humans are unlikely to experience more than modest CO₂ changes, which the sensitivity of Cx26 covers.

Our prior publications cover the interaction of pH and CO₂ on Cx26 gating. Acidification reduces the CO₂-evoked conductance change in HeLa cells expressing Cx26 (Huckstepp et al., 2010a). For CO₂-dependent ATP release from the caudal chemosensory area, a fall in pH from 7.5 to 7.35 roughly halves the amount of ATP released ((Huckstepp et al., 2010b), Fig 4b). We have expanded the Discussion on pp13 to emphasize this point.

Other point that is not assessed or discussed in the study is if the participation of Cx26 in CO₂ sensing involves its gap-junction and/or pannexin action? Is it possible to test these mechanisms? Next, I will enumerate a variety of issues that could help to improve the manuscript:

Cx26^{DN} will affect GJ CO₂-sensitivity (we now provide Supplementary Figure 10 to document this), we cannot exclude GJ role, but there is plenty of evidence for HC role. We have added discussion of this point on pp 9.

-For the Introduction and Discussion, it would be relevant to mention if Cx26 expression has been previously characterized in the area of interest.

Previous work has not specifically looked at this in the caudal parapyramidal area, but (Huckstepp et al., 2010b) shows extensive Cx26 localisation at the ventral medullary surface, which will encompass this area, and the work of Irene Solomon showed its presence in the VLM (including the RTN) and the midline raphe (Solomon et al., 2001). We have added this to the Introduction pp 3.

-I would like to see, in the results section, a more detailed description of the data. For instance, it would be helpful to see central tendency and variability values, the N and the p of the statistical comparisons.

With respect, we are a bit puzzled by this comment as, within the main paper, the boxplots have the individual data points superimposed and show the central tendency and spread of the points. In addition to this the supplementary data shows the data with mean \pm 95% confidence limits. The two tables added to the supplementary information (see above) will help in making these data more accessible.

The statistical comparisons have been made clear in the text and the relevant figures.

-Since most of the data presented in Fig. 1 is already published, I consider that such figure can be removed.

-It is mentioned that FRET interaction between Cx43 and 26 was "much lower than that of Cx26^{WT}Cx26^{WT} or Cx26^{WT}-Cx26^{DN}", but the Figure 2 shows very similar FRET in all Cx combinations. Thus, the images in such Fig. might not be very representative.

We think Fig 1 is helpful, as possibly not all readers will be familiar with the structural biology underlying the design of Cx26^{DN}. We think Fig 1b helps the reader as it is a replotting of published data to allow direct estimation of the Hill coefficient and show a high degree of cooperativity in the action of CO₂ on Cx26. This makes the dominant negative action we show later plausible. The steepness of the dose-response curve is also relevant to the discussion of the range of inspired CO₂ over which Cx26 contributes to the chemosensory reflex.

With regard to FRET in Fig 2, it is important to look at the top row (Clover fluorescence) and how this brightens with bleaching. This shows that for Cx43-Cx26 combination, Clover does not get much brighter with bleaching of mRuby2, and contrasts with the brightening that occurs with Cx26^{WT}-Cx26^{WT} and Cx26^{WT}-Cx26^{DN}.

-As a matter of interpretation, is it possible to speculate about the stoichiometry of the putative heteromers of Cx26^{WT}-Cx26^{DN}?

The cooperativity shown in Fig 1b would suggest it could be as little as 1 dnCx26 to get effect, however we have no data on this point. Our text does allude to this at the bottom of pp4, and given that we have no further information (or easy way of getting it) we feel that further speculation would be hard to justify.

-Is it possible to correlate the amount of CO₂-induced dye-loading, at the different days tested, with the amount of Cx26DN expressed -by the attached fluorescent protein intensity-? Is there a dramatic change in Cx26DN expression between days 5 and 6? Or of its presence in the cell-membrane?

Unfortunately this is not possible -there are no obvious changes that would give this correlation. This is because dnCx26 is expressed within 24h (possibly mainly in intracellular compartments). It does get to the plasma membrane within 48h. However, the dominant negative effect will depend on several factors: i) the amount of Cx26^{DN} in the plasma membrane; the turnover of endogenous Cx26^{WT}; and the rate of assembly into new hemichannels that incorporate Cx26^{DN}. We find that it takes 5-6 days to achieve its effect in HeLa cells. Quite likely the early presence is largely in intracellular organelles, it then has to get to the membrane (it can achieve this within 2d) but its effect will also depend on the rate of assembly and turnover of native hemichannels.

As we cannot get hard evidence to back up these speculations, we have opted not to modify the paper to discuss these matters.

-For the area of interest, a quantification of the amount of GFAP and Cx26DN co-localized is required. A quantification of the amount of Cx26DN no co-localizing with GFAP is also required. Finally, a quantification of the amount of GFAP no co-localizing with Cx26DN is required too.

We now provide this information in a revised version of Supplementary Figure 3 which shows a micrographs in the coronal plane, and Clover fluorescence in the caudal parapyramidal area and GFAP immunoreactivity. We do not provide an estimate of GFAP+ cells with no Clover expression as this is virtually impossible to gain (GFAP does not clearly stain the outlines of individual cells). Nevertheless, it is clear that from the images Clover is expressed in only a small proportion of GFAP+ cells.

-A clear definition of the infected area in all animals is required, perhaps as a scheme, to have an idea of the area in which Cx26 is influencing the response to CO₂. This scheme can be built by the data collected from the corroborations obtained as follows "Co-ordinates were confirmed by post hoc immunostaining for viral driven expression of fluorophores. Data were only included from mice whose injection sites were within the correct location"

We have added a final figure (Fig 7) giving a diagram of the location of the area in coronal and parasagittal planes.

-An indication of the total amount of animals used for this study would be useful.

This has been added to the methods pp 13.

-Explain the conditions in which "additional analgesia was administered"

If any animal was showing signs of pain after surgery additional analgesia was administered as required. This was only required postoperatively. Pain was assessed through the NC3R's grimace scale and by monitoring activity/behaviour. No animals displaying signs of pain or receiving analgesia were used for plethysmography recordings. We have adjusted the text on pp 17.

-A more detailed description of the "custom built plethysmograph" is required. For instance, the dimensions and characteristics of the plexiglass box. The characteristics and/or model of the pressure transducer.

Details added to Methods (pp 19).

-There are some sentences that require proper support by original references:

--"Breathing is a vital function that maintains the partial pressures of O₂ and CO₂ in arterial blood within the physiological limits. Chemosensory reflexes regulate the frequency and depth of breathing to ensure homeostatic control of blood gases"

Added a reference

--"pH-sensitive K⁺ channels (TASKs and KIRs) are potential transducers"

The references are given in the following sentences

--"a subregion that has been more recently studied and termed the "caudal parapyramidal" area."

The references are in the next sentence

--"Peripheral (mainly carotid body) and central chemoreceptors mediate the CO₂-dependent control of breathing"

The references come a bit later in the paragraph

--"there are two components to the adaptive changes in ventilation - an increase in respiratory frequency, and an increase in tidal volume. These two components combine to increase the rate of ventilation of the lungs: minute ventilation"

-There are some typos that should be corrected

--"(Huckstepp et al., 2010b)"

Two close brackets are needed here

--"(Cairn Research))"

Corrected

--"Physiotemp"

Corrected

-Something is missing in the following sentence:

--"Design and characterization of the and mRuby2 protein fluorophores"

Corrected: Clover missing

-I did not understand the message of the following sentence:

--"The nose bar was adjusted so that Bregma was level to a point 2 mm caudal it"

Inserted this text to clarify pp 17:

To maintain consistent placement of the injection pipette, the intra-aural line was adjusted so that bregma was level to a point on the skull 2 mm caudal to bregma.

Reviewer #4 (Remarks to the Author):

In this manuscript, Van de Weil et al. have investigated the role of Connexin26 (Cx26) in central carbon dioxide sensing. The authors provide compelling evidence that Cx26 is important in determining the central response to CO₂ changes by generating and expressing a dominant negative form of the hemichannel both in cultured cells and in an in vivo model of hypercapnia induced respiration. Interestingly, the authors propose a pH-independent role for CO₂ sensing via direct carbamylation of Cx26. This work is supportive of previous work from this group identifying a role for Cx26 in CO₂ sensing. This is a clearly written paper which contains important and compelling data in an area of significant biological interest (mechanisms of CO₂ sensing in mammals).

I have just two comments / suggestions that the authors may wish to consider.

1) In the introduction, the authors should discuss what (if anything) is known about regulation of Cx26 by factors other than CO₂. Do CO₂ levels alter the response of Cx26 to other stimuli. This possibility should at least be discussed.

Intracellular acidification closes Cx26 hemichannels, removal of extracellular Ca²⁺ opens them. There is reported voltage dependence to hemichannel gating at potentials of -20 mV and positive with the voltage of half maximal opening being +10 mV. These very depolarized membrane potentials required for hemichannel opening are unlikely to occur in glial cells. Our electrophysiological experiments demonstrate CO₂-dependent opening at negative membrane potentials (-40 to -50 mV). We have added this information to the introduction on pp4.

2) Can the authors elaborate on whether they believe the carbamylation of Cx26 to be spontaneous or controlled. Is there evidence for an enzymatic process? Have the authors been able to demonstrate by, for example mass spectrometry that the double negative mutant is not carbamylated?

Carbamylation happens spontaneously. This is well documented in the literature for other proteins e.g. RuBisco (Lundqvist and Schneider, 1991), and beta lactamases (Golemi et al., 2001; Maveyraud et al., 2000). We have now added a brief section to the beginning of the Discussion on the physiological significance of carbamylation and given what is known for other proteins as context, pp7 and 8.

It is well known that Arg cannot be carbamylated, and we have demonstrated that the mutations K125R and R104A individually prevent CO₂ sensitivity in other papers. We have demonstrated by MS that K125 is indeed carbamylated, but this is included in another paper in preparation along with a very high resolution cryoEM structure.

References

- Golemi, D., Maveyraud, L., Vakulenko, S., Samama, J.P., and Mobashery, S. (2001). Critical involvement of a carbamylated lysine in catalytic function of class D beta-lactamases. *Proc Natl Acad Sci U S A* 98, 14280-14285.
- Huckstepp, R.T., Eason, R., Sachdev, A., and Dale, N. (2010a). CO₂-dependent opening of connexin 26 and related beta connexins. *J Physiol* 588, 3921-3931.
- Huckstepp, R.T., id Bihi, R., Eason, R., Spyer, K.M., Dicke, N., Willecke, K., Marina, N., Gourine, A.V., and Dale, N. (2010b). Connexin hemichannel-mediated CO₂-dependent release of ATP in the medulla oblongata contributes to central respiratory chemosensitivity. *J Physiol* 588, 3901-3920.
- Lundqvist, T., and Schneider, G. (1991). Crystal structure of the ternary complex of ribulose-1,5-bisphosphate carboxylase, Mg(II), and activator CO₂ at 2.3-Å resolution. *Biochemistry* 30, 904-908.
- Maveyraud, L., Golemi, D., Kotra, L.P., Tranier, S., Vakulenko, S., Mobashery, S., and Samama, J.P. (2000). Insights into class D beta-lactamases are revealed by the crystal structure of the OXA10 enzyme from *Pseudomonas aeruginosa*. *Structure* 8, 1289-1298.
- Solomon, I.C., Halat, T.J., El-Maghrabi, M.R., and O'Neal, M.H., 3rd (2001). Localization of connexin26 and connexin32 in putative CO(2)-chemosensitive brainstem regions in rat. *Respir Physiol* 129, 101-121.

REVIEWERS' COMMENTS:

Reviewer #1 (Remarks to the Author):

The authors have satisfactorily addressed the issues raised in my original review. Also, their inclusion of supplemental data analyses, further explanations and qualifications in the text, and additional figures have strengthened the manuscript and provided more support for their conclusions about the contributions of CO₂-binding to Cx26 of astrocytes in the caudal parapyramidal area of the medulla to chemosensory regulation of breathing in mammals. I do not have additional suggestions for revision.

Reviewer #2 (Remarks to the Author):

The authors totally address my concerns in a very satisfactory way. I do not have more comments on this very interesting and well performed paper.

Reviewer #3 (Remarks to the Author):

Most of my concerns have been addressed and the manuscript has been substantially improved.

Reviewer #4 (Remarks to the Author):

The authors have addressed my comments.